# Inhibition of the CDK2 and Cyclin A complex leads to autophagic degradation of CDK2 in cancer cells

Jiawei Zhang[1,2,12], Yichao Gan[1,3,12], Hongzhi Li[4], Jie Yin [1,3], Xin He[5], Liming Lin[1,6], Senlin Xu [2,7], Zhipeng Fang [2], Byung-wook Kim[2], Lina Gao[2], Lili Ding[2], Eryun Zhang [2], Xiaoxiao Ma[2], Junfeng Li[8], Ling Li[5], Yang Xu [1,6], David Horne[4], Rongzhen Xu[1,6], Hua Yu[9], Ying Gu[1,3,10,11✉] & Wendong Huang [2,7✉]

Cyclin-dependent kinase 2 (CDK2) complex is significantly over-activated in many cancers. While it makes CDK2 an attractive target for cancer therapy, most inhibitors against CDK2 are ATP competitors that are either nonspecific or highly toxic, and typically fail clinical trials. One alternative approach is to develop non-ATP competitive inhibitors; they disrupt interactions between CDK2 and either its partners or substrates, resulting in specific inhibition of CDK2 activities. In this report, we identify two potential druggable pockets located in the protein-protein interaction interface (PPI) between CDK2 and Cyclin A. To target the potential druggable pockets, we perform a LIVS in silico screening of a library containing 1925 FDA approved drugs. Using this approach, homoharringtonine (HHT) shows high affinity to the PPI and strongly disrupts the interaction between CDK2 and cyclins. Further, we demonstrate that HHT induces autophagic degradation of the CDK2 protein via tripartite motif 21 (Trim21) in cancer cells, which is confirmed in a leukemia mouse model and in human primary leukemia cells. These results thus identify an autophagic degradation mechanism of CDK2 protein and provide a potential avenue towards treating CDK2-dependent cancers.

[1] Cancer Institute (Key Laboratory of Cancer Prevention and Intervention, China National Ministry of Education), Second Affiliated Hospital, School of Medicine, Zhejiang University, 310009 Hangzhou, China. [2] Molecular and Cellular Biology of Cancer Program & Department of Diabetes Complications and Metabolism, Arthur Riggs Diabetes & Metabolism Research Institute, Beckman Research Institute, City of Hope, Duarte, CA 91010, USA. [3] Institute of Genetics, Zhejiang University and Department of Human Genetics, Zhejiang University School of Medicine, 310058 Hangzhou, Zhejiang, China. [4] Department of Molecular Medicine, Beckman Research Institute, City of Hope, Duarte, CA 91010, USA. [5] Division of Hematopoietic Stem Cell & Leukemia Research, Beckman Research Institute, City of Hope, Duarte, CA 91010, USA. [6] Department of Hematology, Second Affiliated Hospital, School of Medicine, Zhejiang University, 310009 Hangzhou, China. [7] Irell & Manella Graduate School of Biological Sciences, Beckman Research Institute, City of Hope, Duarte, CA 91010, USA. [8] Department of Translational Research & Cellular Therapeutics, Beckman Research Institute of the City of Hope, Duarte, CA 91010, USA. [9] Department of Immuno-Oncology, Beckman Research Institute of the City of Hope, Duarte, CA 91010, USA. [10] Zhejiang Provincial Key Lab of Genetic and Developmental Disorder, 310058 Hangzhou, Zhejiang, China. [11] Liangzhu Laboratory, Zhejiang University Medical Center, 311121 Hangzhou, Zhejiang, China. [12] These authors contributed equally: Jiawei Zhang, Yichao Gan. ✉email: guyinghz@zju.edu.cn; whuang@coh.org

Cyclin-dependent kinase 2 (CDK2) is best known for the key role it plays during cell cycle progression. This member of the cyclin-dependent kinase (CDK) family is involved in DNA synthesis, G1/S phase transition, and G2 progression modulation[1]. While the monomeric form of CDK2 is inactive, the kinase becomes active, like many other CDKs, when it forms a functional, heterodimeric complex with one of its two regulatory partners – Cyclins A and E. CDK2 modulates a variety of oncogenic signaling pathways when it governs the phosphorylation of a wide range of transcription factors: SMAD3, FOXM1, FOXO1, ID2, as well as UBF, NFY, B-MYB, and MYC[2–8]. Unsurprisingly, the aberrant activation of CDK2, which occurs in many human cancers, leads to uncontrolled cell proliferation during oncogenesis[9]. Likewise, oncogenic processes of several cancers also have been associated with high levels of the two regulatory subunits of CDK2, cyclins A and E[10,11]. Altogether, both the activity of CDK2 and that of its regulatory subunits appear to be important components of oncogenesis.

Although CDK2 is an attractive target to treat cancer, its utility within a clinical setting has been impeded by two major hurdles. First, the mechanism of CDK2 degradation remain poorly understood. Second, many inhibitors developed over the last two decades lack the requisite specificity for CDK2. In general, CDK2 inhibitors can be classified into two groups by their binding sites: inhibitors are either ATP-competitive or non-ATP-competitive. A major problem for the ATP-competitive group is the high sequence homology within ATP-binding sites of different CDKs. The similarities between the binding sites render most ATP-competitive inhibitors either poorly specific or highly toxic[12]. Since these inhibitors have typically failed in clinical trials, none has been approved for commercial use as an anti-cancer drug. In contrast, non-ATP competitive inhibitors target interactions between the CDK2 complex and its substrates. Thus, non-ATP competitive inhibitors tend to have both higher specificity and better efficacy. For example, an in vitro study assessed two short peptides known as Spa310 and CIP; these non-ATP competitive compounds were highly effective inhibitors of CDK2[13].

The activity of CDK2 has been robustly associated with its ability to interact with either Cyclin A or Cyclin E. Disrupting protein-protein interactions (PPIs) between CDK2 and its regulatory cyclins may open a new avenue towards both highly effective and specific inhibitors. Here, we focused on identifying compounds capable of inhibiting the PPI between CDK2 and Cyclin A. To this end, we implemented an in-house developed LIVS (LIgand Virtual Screening) pipeline to screen, in silico, 1,925 FDA-approved drug molecules. According to the results, homoharringtonine (HHT) acted as a potent "disrupter" of PPIs between CDK2 and cyclin A. We thereafter used HHT as a pharmacological tool to study how it targets CDK2. Our results identify an autophagic degradation mechanism of the CDK2 protein in cancer cells.

## Results

**Identification of a small molecule disruptor of the CDK2 complex.** To evaluate whether the PPI interface between CDK2 and Cyclins was druggable, we performed cavity analysis and pocket detection based on CDK2/Cyclin A complex (PDB code: 1FIN)[14]. Five elements comprise the centerpiece of the interface between Cyclin A and CDK2: a C-helix (PSTAIRE helix), activation segment (T-loop), portions of the N-terminal sheet, portions of the C-terminal lobe from CDK2, and the cyclin box from cyclin A[15] (Fig. 1A). Apart from the known ATP-binding pocket in CDK2, we found two druggable pockets (druggable pocket 1 and druggable pocket 2) located in the PPI interface between CDK2 and Cyclin A, which were adjacent to the activation

segment of CDK2 (Fig. 1A). And both druggable pocket 1 and druggable pocket 2 showed a higher Drug Score compared to the ATP-binding pocket (druggable pocket 3) according to the CavityPlus software (Table S1). Therefore, the PPI interface between CDK2 and Cyclin A could be a specific druggable domain.

To predict small molecules that directly interact with druggable pocket 1 or druggable pocket 2 in PPI interface and disrupt the interactions between CDK2 and its regulatory partners (such as Cyclins A and E), we then implemented our in-house developed LIVS pipeline to screen, in silico, 1,925 FDA-approved drug molecules based on CDK2/Cyclin A complex (PDB code: 1FIN). The screen was focused on predicting which compounds were capable of binding to one of the two pockets in PPI interface between CDK2 and Cyclin A (Fig. 1B). From the in silico results, we selected ten top candidates to further validate experimentally (Table S2).

The ability of these candidates (Table S2) to bind CDK2 was then tested in vitro; we used a Drug Affinity Responsive Target Stability (DARTS) assay with whole protein lysate from HEK293 cells[16]. According to the results, homoharringtonine (HHT) significantly protected the CDK2 protein from protease digestion (pronase) (Figs. 1C and S1). The effect, which was dose dependent, suggests physical binding between HHT and CDK2. To further verify a direct interaction between HHT and CDK2, a standard pull-down assay was performed by HHT-conjugated magnetic beads from the whole protein lysate (Fig. 1D). Additionally, when HHT was tested in a competitive pull-down assay, the free compound dramatically inhibited binding between the HHT-labeled beads and CDK2 (as compared to vehicle control, Fig. 1E). We then performed a control to determine whether the binding occurs through the ATP-binding site of CDK2. While this domain is effectively bound by a compound called roscovitine, the compound did not affect interactions between our HHT-beads and CDK2. The results suggest that HHT does not act through the ATP-binding domain. According to our all-around docking model, which was predicted with our in-house developed AAD program, HHT preferentially binds to the PPI site of CDK2. There it can form hydrogen-bonds with Thr47, Arg50, and Arg150 (Fig. 1F). Thus, we both constructed and assessed binding of a CDK2 protein with three site mutations: T47A, R50A and R150A (CDK2-3As). In this variant, interaction between HHT and CDK2-3As was abolished (Fig. 1G). Thus, hydrogen-bonds with T47, R50 and R150 appear critical for binding between HHT and CDK2.

CDK2 promotes cell cycle progression cooperating with CDK1, CDK4/CDK6, and respective cyclins. Among these 4 CDK molecules, CDK2 protein shares the highest similarity with CDK1 protein. To further verify the specificity of HHT for CDK2 binding, we chose the CDK1 as the control and assessed the possible interaction between HHT and CDK1 due to the high homology in both primary sequence and protein structure between CDK1 and CDK2. According to our results from both the DARTS assay and pulldown assay by HHT-labeled beads, the interaction between HHT and CDK1 was significantly weaker than the interaction between HHT and CDK2 (Fig S2A, B). To estimate the structural difference between CDK2 and CDK1, we prepared the corresponding subunit structure from the CDK1-CyclinB complex (PDB code: 4Y72) and CDK2-CyclinA complex (PDB code: 1FIN). The subunit structure alignment demonstrated that CDK2 forms a larger interface than CDK1 with its cognate cyclins, which was consistent with a previous study[17]. The most significant difference was observed in the activation segment between CDK1 and CDK2 (Fig. S2C), which is a critical component of PPI interface between CDK2 and cyclin A. Therefore, these results suggest that HHT could specifically bind

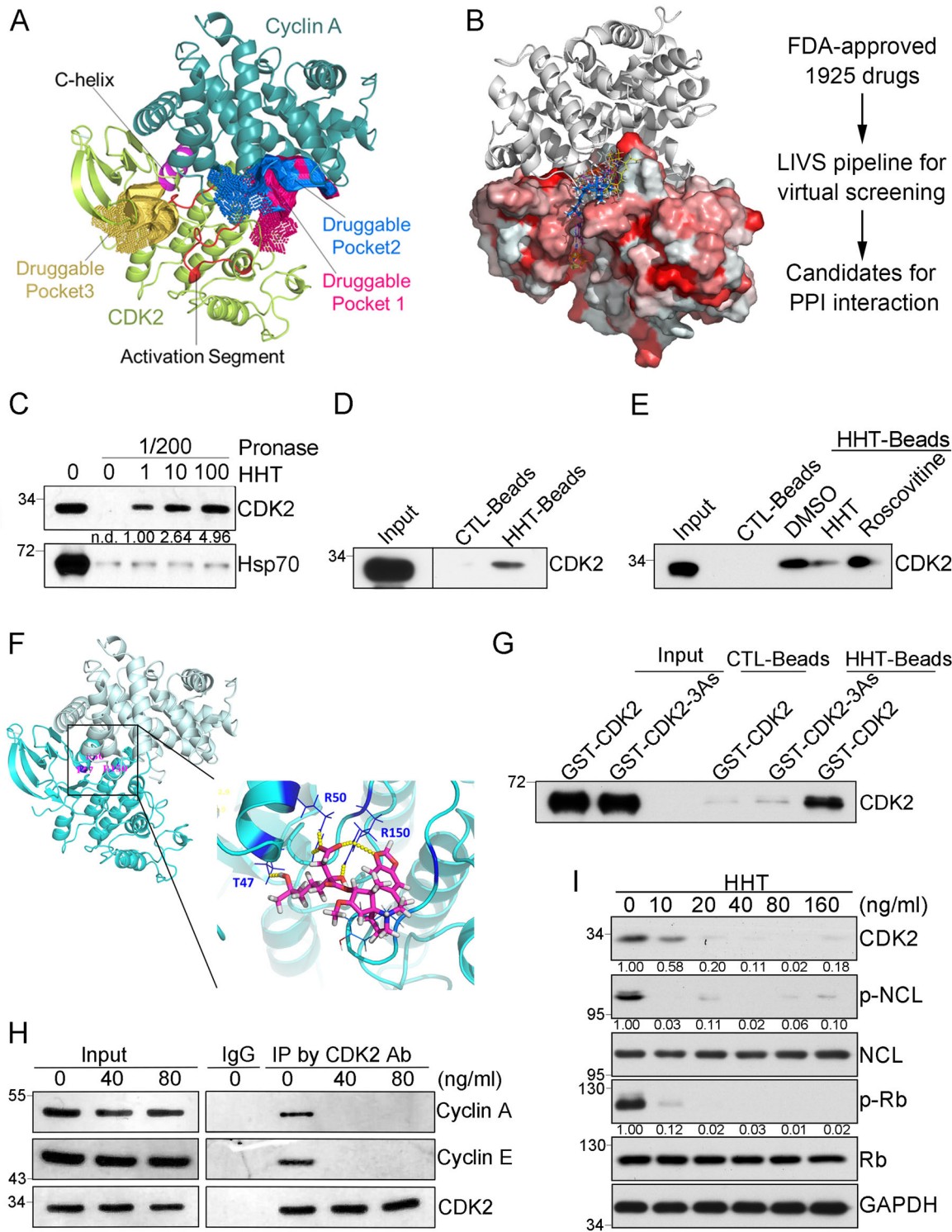

to CDK2 by directly interacting with the PPI interface, which is unique in CDK2/Cyclin A complex.

HHT is a natural alkaloid from Cephalotaxus harringtonia and has been used for more than 30 years as a single agent to treat chronic myeloid leukemia (CML)[18,19]. Recently, HHT has produced very encouraging clinical outcomes for treating acute myeloid leukemia (AML)[20]. Moreover, when we treated different cancer cell lines with HHT, the results demonstrated significant inhibitory effects on cell proliferation (Fig S3). Among these four solid tumor cell lines, prostate cancer cell line PC3 and cervical carcinoma cell line Hela were more resistant to HHT treatment,

while lung cancer cell line A549 and Hepatocarcinoma cell line Hep3B were as sensitive to HHT treatment as leukemia cell line THP1, and the concentration of HHT applied for A549, Hep3B, and THP1 cells could be achieved in the clinic, which is approximately 40 ng/ml with multiple doses and 25 ng/ml with a single dose[21]. We wondered whether the general growth-inhibitory effect of HHT is due to CDK2 complex disruption by HHT. Thus, THP1 leukemia cells were treated with HHT, and within 3 hours the treatment disrupted interactions between CDK2 and its major partners – Cyclin A and Cyclin E (Fig. 1H). Treatment with HHT for 24 hours also reduced phosphorylation

**Fig. 1 Identification of a disruptor of CDK2 complex. A** Druggability prediction for CDK2/Cyclin A complex by CavityPlus. Three druggable pockets identified from the CDK2-CyclinA complex (PDB code: 1FIN). The cartoon illustrated the structures of CDK2 and Cyclin A, which were mainly colored in lemon and deep teal, respectively. Within the CDK2 subunit, the C-helix (residues: 47–57) and activation segment (residues: 145–172) were highlighted in red and purple, respectively. Surfaces of the highly druggable pockets were colored in deep salmon, marine, light orange. Among them, the pocket highlighted in light orange was the known ATP-binding pocket. **B** Virtual ligand screening procedure based on CDK2/Cyclin A complex structure. The left panel was the CDK2/Cyclin A complex structure and virtual screening compounds. FDA-approved drug molecules from LIVS virtual screening pipeline were shown as yellow lines (HHT is colored blue) on CDK2 surface (hydrophobic surface is displayed as red). Cyclin A protein was displayed as a gray cartoon. **C** DARTS assay to identify the interaction between HHT and CDK2 with the incubation of HHT at 0, 1, 10, 100 μg/mL separately. **D** Pulldown assay by HHT-conjugated magnetic beads. **E** Pulldown assay by HHT-conjugated magnetic beads with the co-incubation of either DMSO, HHT (1 μg/mL), or Rosconvitine (50 μM). **F** Binding mode of HHT with CDK2. HHT forms hydrogen bonds (yellow dots) with T47, R50, and R150 in CDK2. **G** Pulldown assay by HHT-conjugated magnetic beads with purified GST-CDK2 or GST-CDK2-3As proteins. **H** Co-immunoprecipitation by CDK2 antibody after THP1 cells treated with different doses of HHT as indicated for 3 hours. **I** Representative western blots of indicated proteins in THP1 cells after HHT treatment for 24 hours. All the western-blotting results shown here were representative of three independent experiments. Source data are provided as a Source data file.

levels of two CDK2 substrates, Rb and Nucleolin (NCL)[22] in a dose-dependent manner (Fig. 1I). These results suggest that treatment with HHT may impair the kinase activity of CDK2.

**CDK2 is critical for AML proliferation and HHT induces CDK2 protein degradation.** Although HHT has been approved for leukemia treatment, the molecular mechanisms behind it are still unclear. It triggered us to study the role of CDK2 in HHT treatment for leukemia. Therefore, we assessed the expression level of CDK2 based on RNA-seq datasets and found that CDK2 mRNA expression level increased significantly in AML cohort in comparison with whole blood samples in normal cohort ($p < 0.0001$) (Figs. 2A and S4A). Meanwhile, we also observed that the mRNA expression level of CDK2 was also significantly increased in recurred samples relative to the corresponding primary samples ($n = 31$, $p < 0.05$) (Figs. 2B and S4B). The analysis indicates that CDK2 expression level is positively correlated with leukemia development. To further determine the role of CDK2 in leukemia cells, we used a CRISPR-Cas9 approach to establish a CDK2 deficient, THP1 cell line (THP1 $^{CDK2-/-}$) (Fig S5A). Compared to knockout (KO) cells, wild-type THP1 cells displayed more rapid proliferation (Fig. 2C). After CDK2 knockout, the cells were more resistant to HHT treatment (Fig. 2D). Further, the cell apoptosis was determined in both THP1 and THP1 $^{CDK2-/-}$ cells after HHT treatment by the Annexin-V/DAPI staining assay. The results showed that both cells had increased Annexin V positive population after 24 hours treatment compared to negative controls, but the increased apoptosis was not enough to explain the effect of HHT on leukemia cell proliferation, and there was no difference between THP1 and THP1 $^{CDK2-/-}$ cells (Fig S5B). These results indicate that growth arrest by inhibiting CDK2 may be a substantial effect of HHT on leukemia cells. We then introduced either exogenous wild-type (WT) CDK2 or mutant CDK2-3As in the THP1 $^{CDK2-/-}$ cells by lentivirus infection, named as THP1 CDK2-wt cells and THP1 CDK2-3As cells, respectively (Fig S5A). Compared to THP1 CDK2-wt cells, THP1 CDK2-3As cells proliferated much more slowly (Fig S5C), which might be due to the impaired activity of CDK2-3As kinase (Fig S5D). But THP1 CDK2-3As cells were more resistant to HHT treatment (Fig S5E). Due to impaired kinase activity of CDK2-3As, we introduced a kinase active mutation at Thr160 (T160E) in CDK2-3As to mimic the phosphorylation of Thr160, which is located in the activation loop of CDK2 and its phosphorylation is critical for the kinase activity[23,24]. We named this CDK2 mutant as CDK2-3As/T160E. In contrast to CDK2-3As, the CDK2-3As/T160E exhibited normal kinase activity on substrate-NCL (Fig S5D), which was comparable to that of wild-type CDK2. The proliferation rate

between THP1 CDK2-3As/T160E and THP1 CDK2-wt cells was similar in the first 4 days, though THP1 CDK2-wt cells demonstrated rapid proliferation on day 5 (Fig. 2E), which suggested that CDK2-3As/T160E could not execute the full activities of wild-type CDK2. As expected, compared to THP1 CDK2-wt cells, the THP1 CDK2-3As/T160E cells showed highly resistant to HHT treatment (Fig. 2F).

The results from different CDK2 mutants suggested that HHT could bind to CDK2 protein via residues T47, R50 and R150, and inhibited its kinase activity. However, if the CDK2 mutants could not fold properly, it would lead to similar results. To determine the consequences of these mutations on the structure, folding and stability of CDK2 protein, we applied the differential scanning fluorimetry (DSF) approach to measure protein thermal unfolding of wild-type CDK2, CDK2-3As, and CDK2-3As/T160E. The results showed that the DSF profiles of wild-type CDK2 (black line), CDK2-3As (red line), and CDK2-3As/T160E (blue line) protein exhibited very similar nanoDSF traces (Fig. S6A). The apparent Tm of wild-type CDK2 was ~51 °C, and the mutants CDK2-3As and CDK2-3As/T160E showed similar thermal stability to the wild-type CDK2 within ~0.2 °C difference. This experiment suggested that mutations of residues T47, R50, and R150 did not affect the folding of CDK2 protein. Further, we compared the kinase activity among CDK2, CDK2-3As and CDK2-3As/T160E by CDK2 kinase assay (Fig. S6B). With the addition of Cyclin A2, the kinase activity of CDK2-3As or CDK2-3As/T160E was lower than that of wild-type CDK2 significantly. The relative activity of CDK2-3As was around 45% and CDK2-3As/T160E was approximately 80% compared to that of wild-type CDK2. When HHT was introduced, the activity of wild-type CDK2 was significantly impaired, while the activity of CDK2-3As and CDK2-3As/T160E did not change much. These results indicated that the HHT could inhibit the in vitro kinase activity of CDK2 via interaction with the residues T47, R50, and R150.

To investigate the effects of CDK2 inhibition in AML in vivo, the THP1 and THP1$^{CDK2-/-}$ cells were stably expressed with luciferase by lentivirus infection and inoculated into NSG mice by tail vein injection. Three days after inoculation, these mice were treated with 0.5 mg/kg of HHT or vehicle intraperitoneally once a day for 2 weeks, and after 2-week break, these mice were treated with 0.5 mg/kg of HHT or vehicle intraperitoneally once a day for 2 weeks (Fig S7A). The experiment demonstrated that CDK2 knockout led to a significant reduction in leukemia cell proliferation and prolongation of survival of THP1 xenograft mice; meanwhile, HHT treatment also inhibited the leukemia development in THP1 xenograft mice but barely had effect on THP1 $^{CDK2-/-}$ xenograft mice (Fig S7B, C). These experiments suggest that CDK2 protein is critical for leukemia cell proliferation and HHT inhibited AML cell proliferation by targeting CDK2.

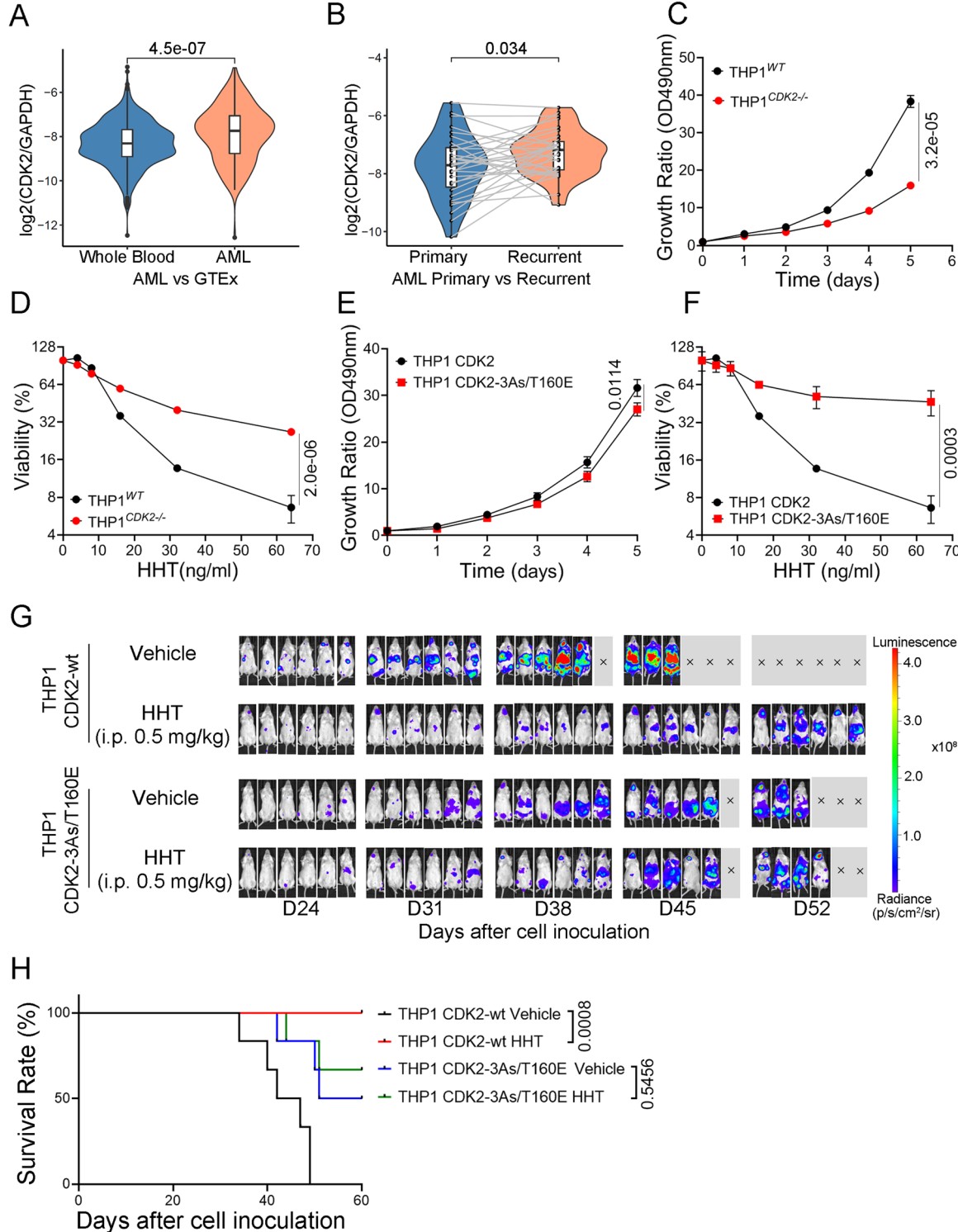

**Fig. 2 CDK2 is critical for AML proliferation. A** Normalized expression level (TPM) of CDK2 relative to GAPDH in TARGET-AML ($n = 156$) and GTEx cohort ($n = 1582$). **B** Normalized expression level (TPM) of CDK2 relative to GAPDH in primary and recurred samples using TARGET-AML ($n = 31$). Box boundaries in **A**, **B** are the 25th and 75th percentiles, the horizontal line across the box is the median, and the whiskers indicate the minimum and maximum values. Wilcoxon Test were used in statistical analysis for **A**, **B**. **C** Growth curves of THP1 wild-type and THP1 CDK2 knockout cells. **D** Cell survival curve of THP1 wild-type and THP1 CDK2 knockout cells after HHT treatment for 24 hours. **E** Growth curves of THP1 CDK2 and THP1 CDK2-3As/T160E cells. **F** Cell survival curve of THP1 CDK2 and THP1 CDK2-3As/T106E cells after HHT treatment for 24 hours. Data presented as mean ± SD from four independent biological samples for each group for **C**–**F**. $P$ values are indicated by two-tailed unpaired Student's $t$ test for **C**–**F**. **G** Bioluminescence imaging of luciferase-expressing THP1 (CDK2-wt or CDK2-3As/T160E) xenograft mice models, treated with 0.5 mg/kg HHT (Six mice for each group). **H** Kaplan–Meier survival of luciferase-expressing THP1(CDK2-wt or CDK2-3As/T160E) xenograft mice models, treated with 0.5 mg/kg HHT or vehicle as the control. $P$ values are calculated by the log-rank test. Source data are provided as a Source data file.

To further elucidate that CDK2 is a critical target of HHT in leukemia treatment, the THP1 CDK2-wt and THP1 CDK2-3As/T160E cells were stably expressed with luciferase by lentivirus infection and inoculated into NSG mice by tail vein injection, and these mice were treated with HHT or vehicle control as described above. At the end of this experiment, all THP1 CDK2-wt xenograft mice with vehicle treatment died, while THP1 CDK2-wt xenograft mice with HHT treatment were all alive. There were 3 mice died in the group of THP1 CDK2-3As/T160E xenograft mice with vehicle treatment and 2 mice died in the group treated with HHT (Fig. 2G, H). This experiment demonstrated that HHT treatment inhibited the leukemia development in THP1 CDK2-wt xenograft mice but had limited effect on THP1 CDK2-3As/T160E xenograft.

When cells were treated with different dose of HHT for 12 hours or with HHT (40 ng/mL) for different time, CDK2 protein was strongly depleted in a dose-dependent and time-dependent manner (Fig. 3A). Since HHT did not affect mRNA levels of CDK2 (Fig S8), we hypothesized that the compound could alter the stability of the CDK2 protein. We assessed whether the half-life of CDK2 protein would be reduced after cells were treated with HHT. First, a translation inhibitor called cycloheximide (CHX) was added to THP1 cells for 1 hour; then cells were treated with either HHT or vehicle control. Protein levels were evaluated at different time points. Without HHT treatment, the half-life of CDK2 protein in cells was 16.59 hours. With HHT treatment, the half-life was shortened to 4.48 hours (Fig. 3B). The half-life of CDK2 protein was also measured with a pulse-chase assay using HaloTag (HT) and HaloTag-tetramethylrhodamine (TMR) ligand systems (Fig. 3C, D). While HHT strongly decreased the half-life of HT-CDK2 fusion protein from 16.11 hours to 5.21 hours, the half-life of mutated HT-CDK2-3As fusion protein was not affected (Fig. 3E). This observation was also repeated in two of our established cell lines, THP1 wild-type CDK2 and THP1 CDK2-3As cells. While wild-type CDK2 protein was decreased after cells were treated with HHT, mutated CDK2-3As protein was unaffected (Fig. 3F). Altogether, HHT appeared to strongly induce degradation of the CDK2 protein.

Previously, a potential mechanism was reported that HHT could inhibit protein synthesis by binding to the A site of ribosome[25]. Our experiments here demonstrated that HHT could inhibited leukemia proliferation in vitro and in vivo by inducing the degradation of CDK2 protein. To test whether CDK2 status might affect protein translation, we determined the mRNA and protein levels of DDX5, STMN1 and MCL1 with protein half-life of ~9.6 hours, ~3.7 hours, and ~30 min respectively, in THP1 cells with different CDK2 status. The results demonstrated that the mRNA and protein levels of DDX5, STMN1 and MCL1 did not change much in these cells (Fig. S9A, B). To further determine whether CDK2 status affects the general translation efficiency, the ribosome profiling was applied into these cells and the results showed that the absorbance peaks of the 40S and 60S ribosome subunits, as well as the monosome and polysome were similar among THP1 cells with different CDK2 status (Fig. S9C), suggesting that CDK2 knockout or CDK2 mutation did not significantly affect the mRNA translation. Combined with the above results, targeting CDK2 may be an independent mechanism underlying antileukemia effect of HHT, which is distinct from inhibiting protein translation.

**A positive correlation between CDK2 protein levels and HHT therapeutic effects**. To determine whether HHT achieves its therapeutic efficacy when it targets the CDK2 protein, we tested the compound in one mouse model of leukemia involving bone marrow (BM) transplantation (BMT) (Fig. 4A). Lethally irradiated C57BL/6 mice (CD45.1) were transplanted with CD45.2 hematopoietic progenitors. The retroviruses that were infected into these progenitor cells drive the expression of an oncogenic fusion gene called MLL-AF9. Ten days post transplantation, MLL-AF9-induced leukemic mice were intraperitoneally injected with HHT (1.0 mg/kg body weight) once daily. The kinetics of tumorigenesis was much slower in the HHT treatment group than in the vehicle group. With HHT treatment, we observed a significantly slower increase in the frequency of CD45.2 positive cells (Fig. 4B). Moreover, mice that received vehicle succumbed to the disease much faster than mice that received HHT (Fig. 4C). As expected, treatment with HHT strongly decreased levels of CDK2 protein in leukemia mice (Fig. 4D). Altogether, the ability of HHT to effectively inhibit the development of leukemia in a mouse model may be relevant to the degradation of CDK2 protein.

To further assess the relationship between treatment with HHT and the CDK2 protein levels, we probed for a significant correlation between the two primary leukemia cells taken from 6 AML patients. These patients were selected to represent different AML subtypes. Both this set of primary leukemia cells and cells from a healthy donor were treated either with HHT or vehicle control; we then measured levels of CDK2. Without treatment, cells from the healthy donor had lower levels of CDK2 protein than cells from all the AML patients (Fig. 4E). Treating the AML cells with HHT reduced levels of the CDK2 protein (Fig. 4F). This suggests a positive correlation between CDK2 protein levels and HHT sensitivity (Fig. 4G). We then evaluated the elevated expression level of CDK2 with overall survival in AML patients, and AML patients ($n = 156$, $p = 0.007$) that express higher CDK2 show significantly poorer overall survival (Figs. 4H and S10).

**Autophagy-lysosome pathway is responsible for CDK2 degradation in cancer cells**. In eukaryotic cells, a vital component of proteostasis is the machinery encompassing heat shock protein 90 (Hsp90) chaperone[26]. Both Hsp90 and its co-chaperone, Cdc37, appear to critically regulate the stability of CDK2, CDK4, and CDK7[27,28]. Therefore, we tested how interactions between Hsp90 machinery and CDK2 were affected when THP1 cells were treated with HHT. To assess this with an anti-FLAG co-IP experiment, CDK2 was both tagged with FLAG (3xFLAG-CDK2) and stably expressed in THP1 cells. When the cells were treated with 40 ng/mL of HHT for three hours, interactions were rapidly disrupted between CDK2 and both Hsp90 and Cdc37 (Fig. 5A). Disrupting these interactions may be responsible for the degradation of CDK2 protein. We further tested this hypothesis with a specific Hsp90 inhibitor called 17-AAG. When cells were treated with 1 μM of 17-AAG, levels of wild-type CDK2 protein decreased in a time-dependent manner (Fig. 5B). Thus, CDK2 may be degraded when either Hsp90 activity is inhibited or the interaction between CDK2 and Hsp90 is disrupted. Both results reinforce that Hsp90 is critical to the stability of the CDK2 protein.

Cellular homeostasis is regulated by two major systems of protein degradation: the ubiquitin–proteasome system (UPS) and the autophagy-lysosome pathway (ALP)[29]. We asked which pathway was involved when HHT induced the degradation of CDK2 protein. THP1 cells were first treated with either MG132 (a proteasome inhibitor, 5 μM) or NH4Cl (an autophagy inhibitor, 20 mM) for 2 hours. Then they received HHT treatment for 9 hours. While NH4Cl significantly blocked the ability of HHT to induce the degradation of CDK2 protein, MG132 did not affect HHT-induced degradation (Fig. 5C upper panel). To exclude the off-targeting

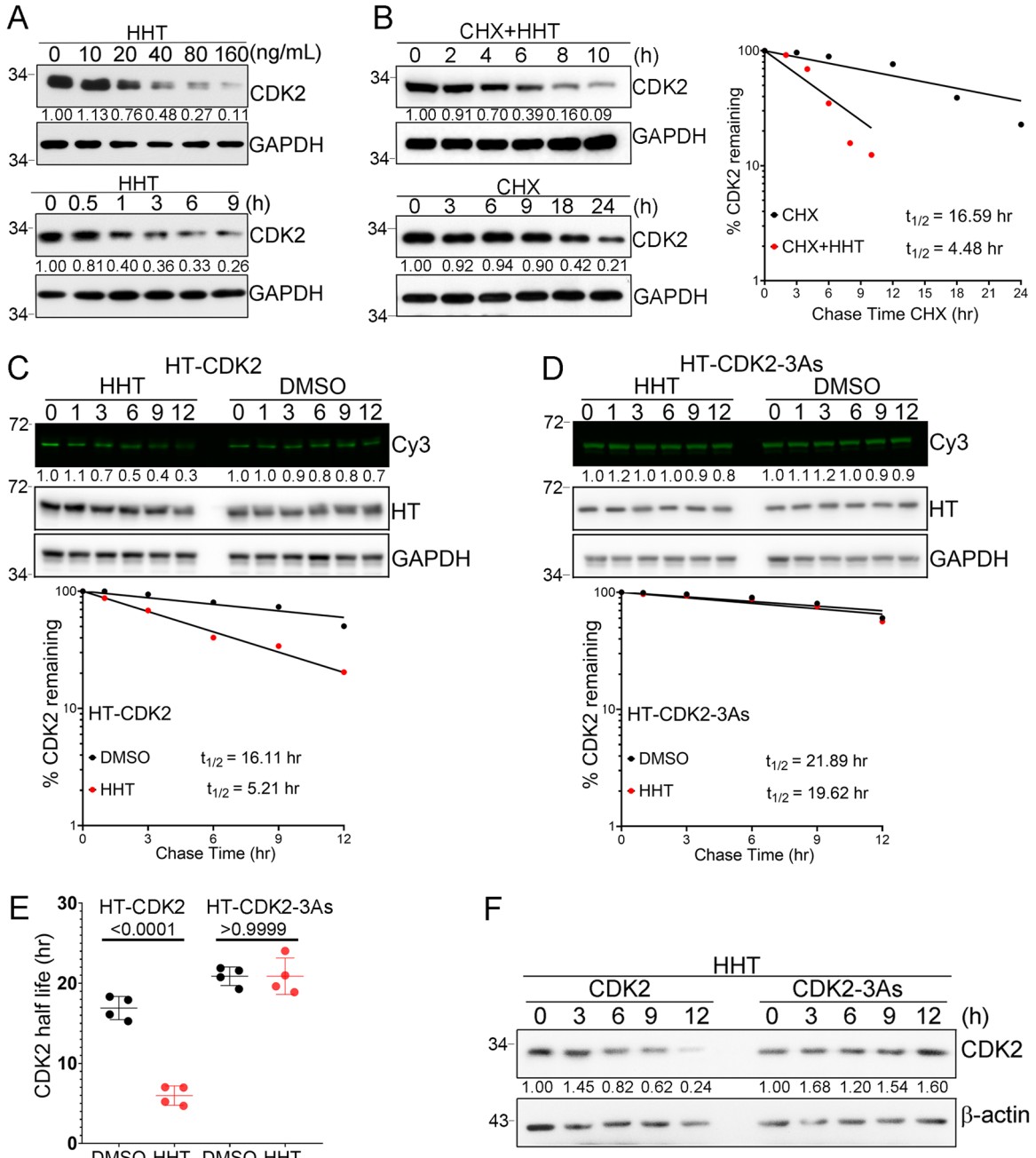

**Fig. 3 CDK2 protein degradation in cancer cells induced by HHT treatment. A** Representative western blots for CDK2 and GAPDH protein levels with HHT treatment. **B** THP1 cells were treated with either CHX (50 μM) alone or CHX plus HHT (40 ng/mL) for the indicated times. The CDK2 and GAPDH protein levels were analyzed by western blot (left panel). CDK2 levels were quantified relative to GAPDH levels and graphed as the percent CDK2 protein remaining after treatment (right panel). Half-lives of CDK2 and CDK2-3As were calculated from exponential line equations and shown as indicated. **C** Pulse-chase analysis of exogenous CDK2 protein using HT-TMR system in 293T cells transfected with HT-CDK2. After labeled with HT-TMR ligand, the cells were treated with 50 ng/ml of HHT for indicated time and subjected to the following experiment. The HT-TMR ligand-labeled HT-CDK2 was visualized with a fluoro-image analyzer (upper panel) and the total expression of HT-CDK2 was analyzed by western blot with GAPDH as the loading control. The relative amount of HT-TMR ligand-labeled HT-CDK2 was measured with densitometric intensity of each band (lower panel). **D** Pulse-chase analysis of exogenous CDK2-3As protein using HT-TMR system in 293T cells transfected with HT-CDK2-3As. After labeled with HT-TMR ligand, the cells were treated with 50 ng/ml of HHT for indicated time and subjected to the following experiment. The HT-TMR ligand-labeled HT-CDK2-3As was visualized with a fluoro-image analyzer (upper panel) and the total expression of HT-CDK2-3As was analyzed by western blot with GAPDH as the loading control. The relative amount of HT-TMR ligand-labeled HT-CDK2-3As was measured with densitometric intensity of each band (lower panel). **E** Summary of exogenous CDK2 and CDK2-3As half-lives in THP-1 cells with or without HHT treatment. Data represent the mean ± SD from four independent biological samples for each group. *P* values are indicated by two-tailed unpaired Student's *t* test. **F** Representative western blot of CDK2 and CDK2-3As protein levels in THP1 cells treated with HHT (40 ng/mL) for the indicated time. All the results shown here were representative of three independent experiments. Source data are provided as a Source data file.

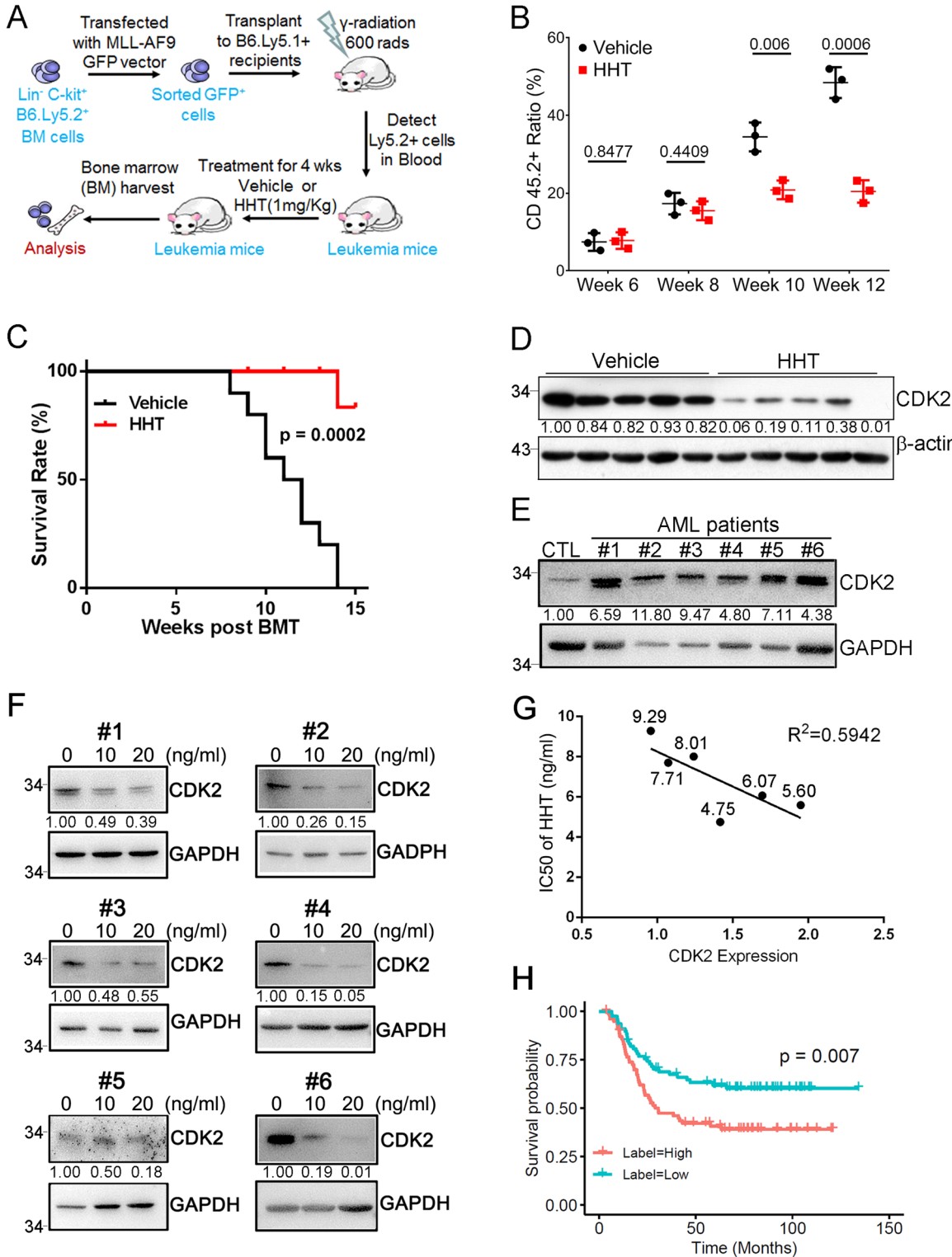

possibility of these two inhibitors, we introduced two more inhibitors, Chloroquine (an autophagy inhibitor, $50\,\mu M$) and lactacystin (a proteasome inhibitor, $10\,\mu M$), to further confirm that the blockage of autophagy pathway but not proteasome pathway could reverse the effect of HHT on the degradation of CDK2 protein (Fig. 5C lower panel). Moreover, when we knock-downed ATG7 by siRNA in THP1 cells, the degradation of CDK2 protein induced by HHT treatment was blocked (Fig. 5D). According to these results, we concluded that the ALP may be mainly responsible for the HHT-induced CDK2 degradation.

**Trim21 recruits autophagy machinery to CDK2 for autophagic degradation.** We next investigated how the degradation of CDK2 protein was mediated by components of the ALP. We assessed this with an anti-FLAG co-IP experiment; CDK2 was both tagged with FLAG (3xFLAG-CDK2) and stably expressed in HEK293 cells. After the cells were treated with either HHT or vehicle control, they were assessed by tandem mass spectrometry (MS/MS). On the SDS–PAGE gel, we detected a specific band in the HHT treatment lane (Fig. 5E). According to our analysis with MS/MS (supplementary Dataset), the effect of HHT could be

**Fig. 4 Induction of CDK2 degradation in vivo. A** Scheme of the MLL-AF9-induced leukemogenesis using a bone marrow transplantation (BMT) assay and pharmacological inhibition by HHT. In these experiments, C57BL/6 (CD45.1) mice were transplanted with hematopoietic progenitors (CD45.2) infected with retroviruses driving the expression of an oncogenic MLL-AF9. When the recipient mice developed leukemia, they received either HHT or vehicle. **B** Flow-cytometry analysis of leukemic cells (CD45.2+) in peripheral blood of leukemic mice treated as indicated at indicated time points after BMT. Data represent the mean ± SD from three independent biological samples for each group. *P* values are indicated by two-tailed unpaired Student's *t* test. **C** Kaplan–Meier survival curves of mice with MLL-AF9-induced leukemogenesis treated as indicated (*n* = 10 mice per group). The *p* value was calculated by log-rank test. **D** Representative western blots of both CDK2 (upper panel) and GAPDH (lower panel) levels in bone marrow cells of the indicated mice treated with either HHT or vehicle for 10 weeks. GAPDH was used as a loading control. **E** Representative western blots of CDK2 expression in BMCs of indicated human primary AML specimens and a healthy donor sample. GAPDH was used as a loading control. Mononuclear cells isolated from primary AML patients and a healthy donor was used for the assay. **F** Western blot analysis of CDK2 expression in BMCs of indicated human primary AML specimens (**E**) treated with or without HHT as indicated. **G** The correlation between CDK2 expression and $IC_{50}$ of HHT in human primary AML specimens (**E**). Correlation is shown using r2 and significance was determined using a Spearman correlation. The numbers close to the dots are the $IC_{50}$ of HHT for each sample (ng/mL). **H** The overall survival of CDK2 for human AML patients with high (red, *n* = 78) versus low (blue, *n* = 78). The patients were dichotomized based on the median value of CDK2 mRNA expression. Wilcoxon Test was used to estimate difference in expression level, and Log Rank test was used in survival analysis. All the results shown here were representative of three independent experiments. Source data are provided as a Source data file.

mediated by three potential candidates: Trim21, Ruvbl2 and Dnaja1. In subsequent anti-FLAG co-IP experiments, HHT appeared to enhance the interaction between CDK2 and Trim21. However, no effects were observed between CDK2 and either Ruvbl2 or DNAJA1 (Fig. 5F). In another co-IP experiment with anti-Trim21 antibody, HHT appeared to enhance the interaction between CDK2 and Trim21 (Fig. 5G). Taken altogether, the interface between CDK2 degradation and the ALP may involve Trim21.

According to previous studies, a subset of tripartite motif (TRIM) proteins act as specialized receptors for highly specific autophagy (precision autophagy). For example, Trim21 both directly binds to its cargo, IRF3, and recruits the autophagic machinery to execute degradation[30]. To further probe the interaction between CDK2 and Trim21, we removed (by deletion mutation) the SPRY domain of Trim21 (Trim21-ΔC) (Fig. 5H). Since the SPRY domain is responsible for binding between Trim21 and its cargoes[30], Trim21-ΔC cannot transport cargoes to autophagic machinery. We tested the impact of Trim21 functionality through co-IP experiments using the anti-FLAG antibody. Prior to being treated with HHT, cells were transfected with both 3xFLAG-CDK2 and either Trim21-HA or Trim21-ΔC-HA. The interaction between Trim21 and CDK2, which was induced by HHT, was abolished when the SPRY domain of Trim21 was deleted (Fig. 5I).

To predict the protein-protein interaction model of CDK2 (3-dimensional structure is extracted from PDB code: 1FIN) and C-terminal region of Trim21 (PDB -code: 2IWG), Zdock software[31] was used to produce 5000 complex models. The best complex model was selected as the one with the best Zdock score, which is also among the largest cluster based on structure similarities of all the models. Moreover, the protein-protein interaction (PPI) residues of Trim21 in complex with IGHG1 protein were also shown on the PPI surface to CDK2, which provided more confidence on the model correctness. As depicted in Fig. 5J, the Trim21 had structural overlaps with cyclin A protein, which demonstrated that Trim21 and cyclin A could not bind to CDK2 simultaneously. It implied that when HHT bound on CDK2 to block the interaction between CDK2 and cyclin A, it might reinforce the binding of free CDK2 with Trim21 to increase the chance of CDK2 degradation. According to this model, the domain located in amino acid residues 171–243 interacted with Trim21 directly, thus we constructed the CDK2 mutant expression plasmid with a deletion of 171–243 amino acid residues. And the cells were transfected with Trim21-HA and 3xFLAG-CDK2 or 3xFLAG-CDK2-Δ171-243. Three hours after HHT treatment, these cells were lysed for co-IP assay by HA-

Beads. And the results demonstrated that deletion of 171–243 amino acid residues in CDK2 protein might abolish the interaction between CDK2 and Trim21 induced by HHT (Fig S11A). However, the deletion of 171–243 amino acid residues in CDK2 may disturb the folding of CDK2 protein, leading to loss of its interaction with Trim21. To exclude this possibility, we further introduced point mutations in CDK2 protein instead of deletion, and based on the prediction of this model that the amino acid residues-V154/P155/M233/P234/P238 could form hydrophobic interactions with amino acid residues in Trim24 to enhance the interaction between CDK2 and Trim21, we constructed the CDK2 mutant expression plasmid with V154R/P155R/M233R/P234R/P238R mutations (CDK2-5Rs). The results of nanoDSF assay demonstrated that the mutated CDK2-5Rs protein shared a similar melting curve with wild-type CDK2 protein (Fig. S6A), indicating that CDK2-5Rs protein has the same folding as wild-type CDK2 protein. Then the cells were transfected with Trim21-HA and 3xFLAG-CDK2 or 3xFLAG-CDK2-5Rs. Three hours after HHT treatment, these cells were lysed for co-IP assay by HA-Beads. And the results showed that V154R/P155R/M233R/P234R/P238R mutations in CDK2 protein, which disrupted the hydrophobic interactions between the two proteins, abolished the interaction between CDK2 and Trim21 induced by HHT (Fig. 5K). Therefore, the half-life of these two CDK2 mutants, CDK2-Δ171-243 and CDK2-5Rs, was not affected by HHT treatment, as demonstrated in pulse-chase analysis of CDK2 mutants using HT-TMR system (Fig S11B). We further determine the interaction between CDK2 and Trim21 in primary AML cells by co-IP assay with Trim21 antibody. As shown in Fig. 5L, HHT treatment enhanced the association of CDK2 and Trim21.

To investigate what role Trim21 might play when HHT induced the degradation of CDK2, THP1 cells were transfected with either Trim21-HA or Trim21-ΔC-HA plasmid; then cells were treated with different dosages of HHT. When Trim21-HA was overexpressed in cells, the rate of CDK2 degradation was faster than the rate in control cells, which harbored HA empty vector (Fig. 6A). However, in cells overexpressing Trim21-ΔC-HA, the rate of CDK2 protein degradation remained similar to control cells (Fig. 6B), due to the loss of interaction between CDK2 and Trim21-ΔC protein. Conversely, when Trim21 was knocked down by siRNA, HHT failed to induce the degradation of CDK2 protein (Fig. 6C). Moreover, when Trim21 was knockdown by siRNA, the degradation of CDK2 protein induced by HHT treatment in primary AML cells was also blocked (Fig. 6D).

We also used a fluorescent imaging approach to probe whether treatment with HHT affected the localization of both CDK2 and

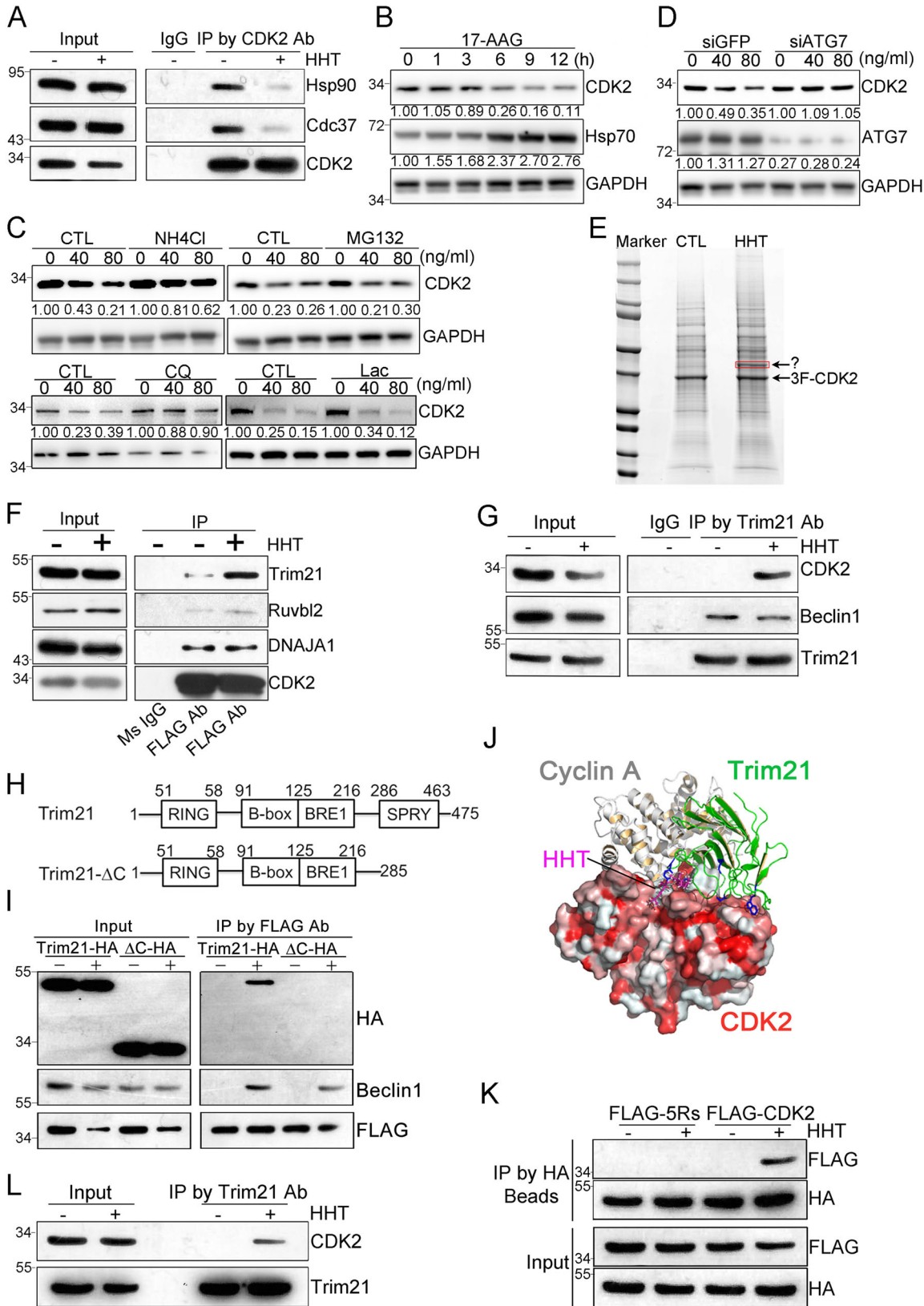

Trim21. To this end, we transiently overexpressed in cells both mRuby2-Trim21 and either CDK2-GFP or CDK2-3As-GFP. After cells were treated with HHT, the localization of both CDK2 and Trim21 was detected with a fluorescence microscope. As shown in Fig. 6E upper panel, the treatment both strongly reduced the levels of CDK2-GFP and induced co-localization between CDK2-GFP and mRuby2-Trim21; these effects were followed by autophagic degradation of CDK2. In contrast, a similar cascade was not observed for CDK2-3As-GFP. We checked the colocalization between CDK2-GFP and mRuby2-Trim21 induced by HHT in three different replicates, and the results showed significant colocalization in CDK2-GFP group but

**Fig. 5 Identification of Trim21 as a mediator of CDK2 autophagic degradation induced by HHT treatment. A** Representative co-immunoprecipitation assay in THP-1 cells treated with HHT (40 ng/mL). **B** Western blot analysis in THP-1 cells treated with 1 μM of 17-AAG. **C** Western blot analysis in THP-1 cells were pretreated with NH4Cl (20 mM) or MG132 (5 μM) (upper panel) and Chloroquine (50 μM) or lactacystin (10 μM) (lower panel), together with HHT treatment for 9 hours. **D** Two days after ATG7 or GFP siRNA transfection, THP-1 cells were treated with HHT for 9 hours and subjected to western blot analysis. **E** HEK293 cells stably expressing 3xFLAG-CDK2 were treated with either HHT (50 ng/mL) or DMSO for 6 hours and proteins were separated by SDS–PAGE. The specific band in the HHT treatment lane (as indicated) was cut for LC/MS analysis. Co-immunoprecipitation assay by CDK2 antibody (**F**) or Trim21 antibody (**G**) in cells either treated with HHT (50 ng/mL) or DMSO for 6 hours, followed by western blot analysis. **H** Trim21 domains and the deletion mutant were constructed as indicated. **I** HEK293 cells stably expressed 3xFLAG-CDK2 were transfected with either Trim21-HA plasmid or Trim21-HA-ΔC plasmid and treated with HHT (50 ng/mL) for 6 hours, followed by co-immunoprecipitation with FLAG-beads and western blot analysis. **J** The complex model of Trim21 and CDK2 proteins. The protein-protein interaction residues on Trim21 (green-colored cartoons) C-terminal region to protein IGHG1 (PDB -code: 2IWG) are displayed as blue-colored sticks. The cyclin A protein is also superimposed as gray cartoons (PDB -code: 1FIN) to show its overlap with Trim21. The model of HHT bound to CDK2 is superimposed as pink sticks. **K** HEK293 cells transfected with Trim21-HA plasmid and 3xFLAG-CDK2 or 3xFLAG-CDK2-5Rs plasmid were treated with HHT (50 ng/mL) for 3 hours, followed by co-immunoprecipitation with HA-beads and western blot analysis. **L** Representative co-immunoprecipitation assay in mononuclear cells isolated from one primary AML patient were treated with HHT (40 ng/mL) or DMSO for 3 hours. Source data are provided as a Source data file.

not in CDK2-3As-GFP group after HHT treatment (Fig. 6E lower panel). Taken altogether, we proposed a putative model in which HHT blocks the interaction between CDK2 and Cyclin A. This enhances interactions between CDK2 and Trim21, which recruits autophagic machinery to degrade CDK2 in cancer cells (Fig. 6F).

## Discussion
Dysregulation of CDK2 and its complex plays a critical role in both the malignant transformation of cells and tumorigenesis. When CDK2 inhibitors are applied to a range of cancer cells in vitro and tumor mice models in vivo, the inhibitors exhibit an encouragingly anti-tumor effect. However, their utility within a clinical setting has been impeded largely due to the lack of requisite specificity for CDK2.

The vast majority of CDK2 inhibitors target the ATP-binding site of CDK2. Six examples are flavopiridol, R547, roscovitine, P276-00, AT7519, and NU2058. These inhibitors failed to meet expectations during clinical trials because they have either poor specificity or high toxicity. One major impediment to developing highly specific, small molecular inhibitors is the high sequence homology within the ATP-binding sites of CDK family members and other kinases[32]. Therefore, instead of targeting the ATP-binding site, emerging studies have used non-ATP competitive inhibitors to disrupt the interaction between CDK2 and either its partners or substrates. Several non-ATP competitive inhibitors of CDK2 have shown promising clinical applications; two examples are a 39-residue peptide called Spa310 and a p53-derived peptide called CIP[32]. However, a major gap in developing the approach is a lack of analogous small molecules. Here, we screened and identified a non-ATP competitive inhibitor of CDK2. This small molecule, called HHT, is an FDA-approved drug. In this study, we showed that it can induce the autophagic degradation of CDK2 and suppress leukemia development. Due to aberrant CDK2 activity in other cancers including ovarian cancer, Hepatocellular carcinoma, glioblastoma, prostate cancer, and B cell lymphoma[9], it is highly possible that HHT could also have the anti-tumor effects in these cancers in addition to leukemia.

We first established a virtual ligand screening method called the LIVS pipeline[33–36]. This multiple-stage program provides full-coverage when we screened our set of virtual ligands. While other computational models typically produce a hit rate (defined as percentage of virtual ligand screening compounds that passed the primary assay in the wet-lab validation with a typical concentration of 20–50 μM) of ~3%[37,38], the hit rate for LIVS is around 15%[33,36]. We achieve this improvement by integrating four key elements: Lipinski's rule of five, HTS frequent hitter (PAINS), protein reactive chemicals as oxidizer or alkylator (ALARM), and maximization of the molecular diversity[34,36].

Using LIVS, 1925 FDA-approved small drug molecules were screened; 10 compounds were selected as the top candidates to target the interaction between CDK2 and cyclin A. Moreover, these ten candidates were verified by DARTS experimentally. Under physiological conditions, this method can detect interactions between small molecules and proteins. By compiling the results from both LIVS and DARTS, we identified one FDA-approved drug which disrupted the CDK2 complex – HHT. While HHT was not active against the CDK1 protein, it specifically inhibited the activity of the CDK2 protein. HHT has been used for more than 30 years as a single agent to treat chronic myeloid leukemia (CML)[18,19]. HHT was recently assessed for patients with de novo AML in a multicenter, open-label, and randomized phase III trial. Compared to a standard regimen that is used to induce remission of de novo AML (a regimen of daunorubicin plus cytarabine), a combination of three drugs, HHT + cytarabine + aclarubicin, achieved a higher complete remission rate of 73%[20]. Previous reports suggest this compound may likely operate through a broad mode of action including protein synthesis inhibition and DNA epigenome modulation[39,40]. However, it is uncertain whether other undefined mechanisms underly the anti-tumor effects of HHT. Here we show that HHT can not only directly bind to the PPI site of CDK2 but also disrupt interactions between CDK2 and its cyclin partners. Moreover, this binding both inhibited the activity of CDK2 and induced its degradation in cancer cells. Such findings thus highlight a potential therapeutic strategy to target CDK2-associated malignancies: the PPI between CDK2 and cyclins can be disrupted by a non-ATP competitive inhibitor of CDK2. Based on these findings, further studies are warranted to analyze the distinct functional groups in HHT molecule, which interact with specific target respectively to fulfill different activities including disruption of CDK2 PPI and protein translation blockage. Moreover, novel HHT derivatives could be developed to inhibit different targets with higher specificity and lower requisite dosage with the analysis of functional groups in HHT molecule.

In mammalian cells, proteins are degraded by either the ubiquitin-proteasome system or the autophagy-lysosome system. While a specific E3 ubiquitin ligase called KLHL6 mediated the ubiquitin-dependent proteasome degradation of CDK2 in leukemia cells[41], in breast cancer cells, β-transducing repeat–containing proteins (βTrCP) controlled the autophagic degradation of CDK1[42]. During the highly conserved mechanism of the autophagy-lysosome system, lysosomes mediate the degradation of both proteins and organelles. The process plays a crucial role in maintaining cellular homeostasis[43]. In many cases, autophagic receptors will first recognize autophagic tags, such as ubiquitin and galectins, and then deliver the tagged cargo for

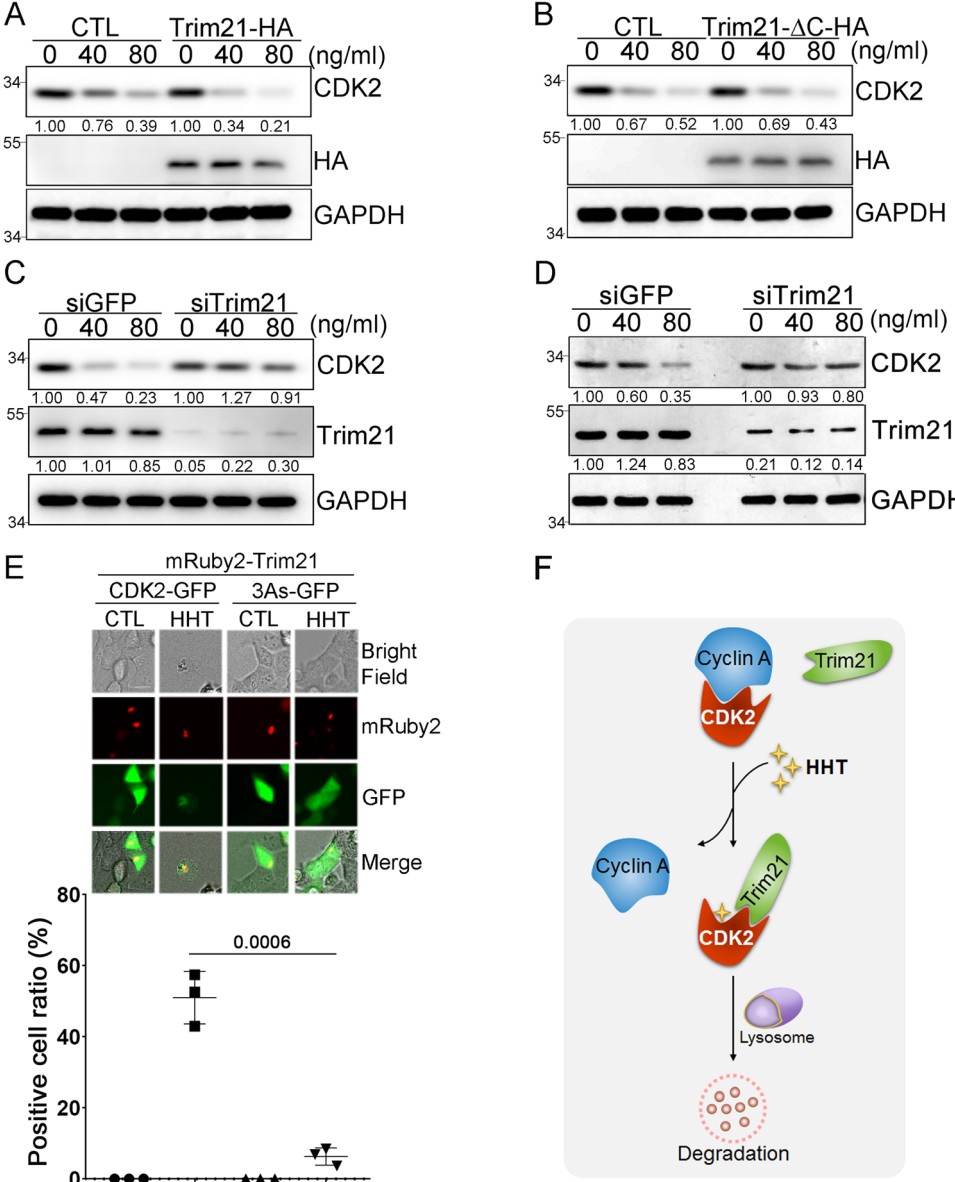

**Fig. 6 Trim21-mediated CDK2 autophagic degradation in cancer cells. A** THP1 cells transfected with either Trim21-HA plasmid or empty vector were treated with HHT as indicated for 9 hours, and the protein levels of CDK2 and Trim21 were analyzed by western blot with β-actin as the loading control. **B** THP1 cells transfected with either Trim21-HA-ΔC plasmid or empty vector were treated with HHT as indicated for 9 hours, and the protein levels of both CDK2 and Trim21 were analyzed by western blot with GAPDH as the loading control. **C** THP1 cells were transfected with siRNA targeting either Trim21 or GFP, and after 48 hours treated with HHT as indicated for 9 hours. The protein levels of CDK2 and Trim21 were analyzed by western blot with GAPDH as the loading control. **D** Mononuclear cells isolated from a primary AML patient were transfected with siRNA targeting either Trim21 or GFP. Forty-eight hours after transfection, cells were treated with HHT as indicated for 12 hours, and the protein levels of CDK2 and Trim21 were analyzed by western blotting with GAPDH as the loading control. **E** HEK293 cells were transfected with either CDK2-GFP or CDK2-3As-GFP as well as mRuby2-Trim21; then cells were either treated with HHT (50 ng/mL) or DMSO for 9 hours, and the localizations of CDK2 (green) and Trim21 (red) were determined by fluorescence microscopy (upper panel). Scale bar, 20 μm. The positive cell ratio with colocalization between CDK2-GFP and mRuby2-Trim21 after HHT treatment in three different experiment replicates was analyzed (lower panel). Data represent the mean ± SD from three independent biological samples for each group. P values are indicated by two-tailed unpaired Student's *t* test. **F** A putative model of Trim21-mediated autophagic degradation of CDK2 protein induced by HHT. All the results shown here were representative of three independent experiments. Source data are provided as a Source data file.

autophagic degradation[44,45]. According to recent studies in autophagy, a subset of TRIM proteins act as both receptors and platforms upon which core regulators of autophagy can be assembled. As autophagic receptors, TRIMs can directly recognize either endogenous or exogenous targets without autophagic tags. These unique features allow TRIMs to govern cargo degradation through a highly exact process termed 'precision autophagy'[30,46].

In this study, when HHT bound to CDK2 protein, the interaction between CDK2 and its partners was disrupted. Subsequent contact with Trim21 appeared to cause the CDK2 protein to degrade through the autophagy-lysosome system. Degradation of CDK2 by HHT was observed in in vitro cell studies, in vivo animal models, and primary patient samples. While only a handful of studies have probed the mechanisms by which CDK2 is degraded[47], the above process may represent a new mechanism

of CDK2 degradation. Although all CDKs appear to be maintained at similar levels throughout the entire length of the cell cycle, different stages of the cell cycle have different rates of both cyclin synthesis and degradation. This in turn provides control over cell cycle progression when these cyclins bind to and regulate the activities of CDKs. In this context, the degradation mechanism we have identified may open a potential avenue towards CDK2-targeted cancer therapies.

## Materials and methods

**Druggability prediction**. We evaluate the cavity and predict druggable pockets without ligand via CavityPlus[14], the cutoff was set as average pKd > 6.5 based on the CDK2-CyclinA complex (PDB code: 1FIN).

**Virtual ligand screening and computational modeling**. As a multiple-stage program for virtual ligand screening, our in-house developed LIVS (LIgand Virtual Screening) pipeline uses the three HTVS/SP/XP precisions of Glide software for docking in series[48]. LIVS provides full-coverage and docks every compound in the library. To control the qualities of output compounds, this automatic program integrates Lipinski's rule of five[49], HTS frequent hitter (PAINS)[50], protein reactive chemicals such as oxidizer or alkylator (ALARM)[51], and maximization of the molecule diversity by using UDScore (Universe Diversity Score, developed by us to measure library diversity which is independent of library size). The CDK2 three-dimensional structure was downloaded from RCSB PDB[52] ID: -1FIN[15]). The 1,925 FDA-approved small drug molecules were downloaded from DrugBank[53]. Ten FDA-approved drugs were selected from LIVS pipeline for validation in the wet lab (Table S2).

The binding mode of HHT was calculated by our in-house developed AAD (All-Around Docking) program[54]. AAD methodology searches the best binding site and binding pose around the whole protein surface without any knowledge of the possible locations of docking pockets. The program predicted that HHT prefers binding to the PPI site compared to the ATP-binding site (Fig. 1E). The structural figures were produced by using PyMOL v1.8.6 software (The PyMOL Molecular Graphics System, Schrödinger, LLC.).

**Drug Affinity Responsive Target Stability for small-molecule target identification**. The DARTS protocol was described in our previous work[16]. Briefly, HEK293 cells were lysed in M-PER buffer with the addition of both protease inhibitors and phosphatase inhibitors. After chilled TNC buffer (50 mM Tris-HCl pH 8.0, 50 mM NaCl, 10 mM CaCl2) was added to the protein lysate, the protein concentration of the lysate was measured by the BCA Protein Assay kit (Pierce, 23227). The protein lysate was then incubated for 3 hours at room temperature with either vehicle control (H2O) or varying concentrations of small molecules as indicated; materials were shaken at 600 r.p.m. in an Eppendorf Thermomixer. Pronase digestions, which were performed for 20 min at room temperature, were stopped by adding SDS loading buffer and immediately heating at 70 °C for 10 min. Samples were both subjected to SDS–PAGE and western blotted for CDK2. Hsp70 was used as a negative control.

**Pulldown assay by HHT-conjugated magnetic beads**. The HHT-conjugated magnetic beads were synthesized as below: one ml of Absolute Mag™ Epoxy Magnetic Particles (#WHM-Q009, CD Bioparticles) was washed with dimethylformamide (DMF) for three times and suspended in 5 ml of DMF. Seventy-five mg of HHT was added into mixture with 200 μL of triethylamine (TEA), and the mixture was stirred overnight at room temperature protected from light. The magnetic beads were washed by PBS for 5 times to remove unreacted residue and resuspended in 1 ml of TBS. After conjugation, fourier transform infrared spectroscopy (FTIR) was used to check the binding between the epoxy of the beads and HHT. HEK293 cells were lysed in M-PER buffer with the addition of both protease inhibitors and phosphatase inhibitors. One hundred microlitre of HHT-conjugated magnetic beads and control magnetic beads were added into the protein lysate separately, and the mixture was rotated at 4 °C overnight. Then the magnetic beads were washed by M-PER buffer for 5 times and the proteins binding to the HHT-conjugated magnetic beads were harvested by heating the samples in 60 μL of SDS loading buffer at 95 °C for 10 min.

**Bacterial expression and purification of recombinant proteins**. Full-length human CDK1 and CDK2 were expressed as a GST-N-terminal tagged fusion protein from pGEX-6P-1 recombinant vector. BL21 (DE3) strain of *E. coli* were transformed and grown in LB medium containing Ampicillin overnight at 37 °C. Then cultures were diluted 1:100 and grown at 25 °C for 2 hours; they were then induced with 0.2 mM isopropyl-b-D-thiogalactoside (IPTG) and further incubated for an additional 6 hours. Bacteria were pelleted at 4000×g for 20 min at 4 °C. The pellets were then suspended in the buffer (50 mM Tris/Cl (pH 8), 250 mM NaCl, 1% Triton X100, 10% Glycerol, and 1 mM PMSF). Lysozyme (1 mg/ml) was then added into the suspension and the suspension was incubated on ice for 30 min, followed by sonication on ice. Insoluble materials were pelleted by centrifugation at

100,000×g for 60 min at 4 °C. GST-tagged proteins were purified on glutathione agarose (Sigma) pre-equilibrated with lysis buffer, washed several times with the same buffer, and finally eluted in 50 mM Tris/Cl (pH 8), 10% glycerol, and 20 mM reduced glutathione. Purified GST-tagged proteins were then dialyzed overnight against the same buffer in the absence of glutathione at 4 °C.

**Murine bone marrow transduction/transplantation model**. The retroviral vector MSCV-IRES-eGFP carrying the fusion gene of MLL-AF9 cDNA was used to make high-titer, helper-free, replication-defective ecotropic virus stock by transient transfection of 293T cells. Six- to eight-week-old female C57BL/6 mice were used as donor mice for tumorigenesis experiments. Bone marrow from 5-fluorouracil (5-FU)-treated (150 mg/kg) donor mice was transduced with MLL-AF9 retrovirus by co-sedimentation in the presence of interleukin-3 (IL-3), IL-6, and stem cell factor. Syngeneic recipient mice were prepared by 1300 cGy g-irradiation; a dose of $0.5 \times 10^6$ transfected cells was transplanted via tail vein injection. After transplantation, recipient mice were evaluated daily for signs of morbidity, weight loss, and failure to thrive.

**CDK2 kinase assay**. CDK2 kinase assay was executed with the CDK2 assay kit (#79599, BPS Bioscience). The protocol is following the manufacturer's protocol described as below: prepare the master mixture (6 μL of 5x Kinase assay buffer 1 + 1 μL of ATP (500 μM) + 5 μL of 10x CDK substrate peptide 1 + 13 μL of distilled water). Add 25 μL of master mixture to every well of the 96-well plate. Add 20 ng of Cyclin A2 and 30 ng of different CDK2 mutant protein into the wells as indicated along with 100 ng/mL of HHT. Incubate at 30 °C for 45 minutes. After the 45-minute reaction, add 50 μL of Kinase-Glo Max reagent to each well. Cover plate with aluminum foil and incubate the plate at room temperature for 15 min. Measure luminescence using the microplate reader. "Blank" value is subtracted from all readings. The relative kinase activity of Cyclin A2/wild-type CDK2 group is set as 100%.

**Immunoprecipitation assay**. Immunoprecipitation by anti-FLAG M2 beads was done using the FLAG immunoprecipitation Kit (Sigma, St. Louis, MO). After treatment, cell protein was extracted with IP lysis buffer (Thermo, Waltham, MA). Cell lysate was then incubated with pre-washed anti-FLAG M2 beads (50 μl) overnight at 4 °C with gentle rotation. Immunoprecipitated complexes were collected by centrifugation (7000×g, 1 min, 4 °C) and washed three times with 1 mL washing buffer by re-suspension and centrifugation (7000×g, 1 min, 4 °C). The immunoprecipitate was detected by western blot analysis.

**AML xenograft model and Xenogen imaging**. All animal procedures were approved by the Institution's Ethics Committee. To establish xenograft model, human AML THP-1 cells with different CDK2 status were stably transduced with a lentiviral firefly luciferase, and then $1 \times 10^5$ cells were injected through the tail vein into NSG (NOD/SCID/IL2Rγ−/−) mice (five- to six-week-old female). Three days after injection, HHT (0.5 mg/kg body weight) or vehicle control was intraperitoneally injected for 14 consecutive days. After two-week break, the mice were treated with the same protocol as above. Bioimaging of mice was performed at different time points using an in vivo IVIS 100 bioluminescence/optical imaging system (Xenogen, Alameda, CA). Briefly, mice were intraperitoneally injected with D-Luciferin (2.5 mg per mouse) (Promega, Madison, WI) dissolved in PBS 15 min before measuring the luminescence signal. General anesthesia was induced with 5% isoflurane and continued during the procedure with 2% isoflurane introduced through a nose cone.

**Peripheral blood mononuclear cells isolation**. Mononuclear cells were isolated using lymphocyte separation medium from the peripheral blood samples of patients with AML and normal volunteers with their written informed consent. Briefly, blood was diluted 1:1 in phosphate-buffered saline (PBS) at RT prior to layering over Lymphocyte Separation Medium. Peripheral blood mononuclear cells were collected following centrifugation (800×g, 20 min) and washed in PBS (320×g, 10 min). Cells were resuspended in RPMI-1640 medium supplemented with 10% fetal bovine serum. This study was carried out in accordance with the recommendations of Ethics and Scientific Committee of The Second Affiliated Hospital of Zhejiang University School of Medicine with written informed consent from all subjects.

**CDK2 knockout by the CRISPR/Cas9 system**. The sgRNA sequences targeting human CDK2 were designed using the CRISPR on-line design tool (www.genome-engineering.org/crispr). The designed sequence was cloned into the pX330 plasmid (Addgene). The 20-nt guiding sequences targeting exon1 and exon2 of human CDK2 are shown below: 5′- aaaggtggaaaagatcggag-3′ (target exon1 for clone A), 5′- gaatctgcttattaacacag-3′ (target exon2 for clone B). The sgRNA-containing pX330 vector was co-transfected into cells with the pMAX-GFP plasmid. After 48 hours, green fluorescent protein positive (GFP+) cells were single-cell sorted by fluorescence-activated cell sorting (FACS) into 96-well plates. Single clones were then expanded and screened by western blot analysis. Genomic DNA was purified

from clones using the QIAamp DNA mini kit and underwent sequencing to verify the deletion.

**Cycloheximide chase assay**. Cells were treated with 50 μM cycloheximide (CHX) and harvested at the indicated time points. Protein was extracted from the cells and subjected to western blot analysis. Protein levels were measured with the densitometric intensity.

**HaloTag pulse-chase assay**. The protein of interest (POI) was cloned into HaloTag (HT) vector to establish a HT-POI vector. Cells (1 million cells in 6-well plates) were transfected with 0.5 mg of HT-POI. After culturing them for 40 hours, cells were first incubated with 5 μM HaloTag-tetramethylrhodamine (TMR) ligand (Promega) for 15 min to allow for pulse labeling of the HT-POI and then washed by PBS twice. After that, the cells were suspended in lysis buffer. The whole cell extract was subjected to SDS–PAGE. The TMR-labeled HT-POI was visualized with a fluoro-image analyzer FLA-3000G (FUJI FILM, Tokyo, Japan).

**Survival/proliferation assay**. The MTS assay, which measures cell survival, was conducted with the CellTiter 96 Aqueous Cell Proliferation Kit (Promega). The $IC_{50}$ was defined as the drug concentration that induced a 50% viability decrease and calculated by GraphPad Prism software.

**siRNA transfection**. The transfection was following Lonza's protocol with Nucleofector I Device. One million cells were harvested and centrifuged at 90×g for 10 min at room temperature. Resuspend the cell pellet carefully in 100 μl room-temperature Nucleofector Solution per sample. Combine 100 μl of cell suspension with 300 nM siRNA. Transfer cell/DNA suspension into certified cuvette and insert the cuvette with cell/DNA suspension into the Nucleofector Cuvette Holder. Select the appropriate Nucleofector Program U-01 and apply the selected program. Take the cuvette out of the holder once the program is finished. Immediately add 500 μl of the pre-equilibrated culture medium to the cuvette and gently transfer the sample into the prepared 12-well plate. Incubate the cells in humidified 37 °C/5% CO2 incubator until analysis. The siRNAs for ATG7 and Trim21 are SMART pool from Dharmacon: ATG7 (#L-020112–00) and Trim21 (#L-006563-00). The siRNA for GFP is 5′-CAGAUGAACUUCAGGGUCAGC-3′.

**Western blot analysis**. Cell specimens were washed twice with PBS buffer; total cellular protein was extracted using Radio-Immunoprecipitation Assay buffer (RIPA). Cell extracts were subjected to sodium dodecyl sulfate–polyacrylamide gel electrophoresis (SDS–PAGE; 10% polyacrylamide gels); then they were transferred to polyvinylidene difluoride (PVDF) membranes (Bio-Rad) and blocked with 5% nonfat milk (Bio-Rad) in TBS–Tween 20 (TBST). The membranes were then reacted with primary antibodies overnight at 4 °C. After 3 washes with TBST, membranes were probed with a horseradish peroxidase–conjugated secondary antibody for 1 hour at room temperature and reacted with SuperSignal West Pico Chemiluminescent Substrate (Thermo). The antibodies were purchased from Proteintech as below: β-actin Monoclonal Antibody (#66009-1-Ig,1:8000); HSP90 Polyclonal Antibody(#13171-1-AP,1:1000); CDC37 Polyclonal Antibody(#10218-1-AP,1:1000); ATG7 Polyclonal Antibody(#10088-2-AP,1:1000); TRIM21 Polyclonal Antibody(#12108-1-AP,1:1000); RUVBL2 Polyclonal Antibody(#10195-1-AP,1:1000); DNAJA1 Polyclonal Antibody(#11713-1-AP,1:1000); Beclin 1 Polyclonal Antibody(#11306-1-AP,1:1000); GAPDH Monoclonal Antibody(#60004-1-Ig,1:8000); HSP70 Polyclonal Antibody(#10995-1-AP,1:1000); Cyclin A2 Polyclonal Antibody(#18202-1-AP,1:1000); GST Tag Monoclonal Antibody(#66001-1-Ig,1:8000); CDK1-Specific Polyclonal Antibody(#19532-1-AP,1:1000). The antibodies were purchased from Cell Signaling Technology as below: DDX5 (D15E10,1:1000) XP® Rabbit mAb(#9877,1:1000); Stathmin Antibody(#3352,1:1000); Mcl-1 (D5V5L) Rabbit mAb(#39224,1:1000); Anti-mouse IgG, HRP-linked Antibody(#7076,1:3000); Anti-rabbit IgG, HRP-linked Antibody(#7074,1:3000); HA-Tag (C29F4) Rabbit mAb(#3724,1:1000); CDK2 (E8J9T) XP® Rabbit mAb(#18048,1:1000); Cyclin E1 (D7T3U) Rabbit mAb(#20808,1:1000); Nucleolin (D4C7O) Rabbit mAb(#14574,1:1000); and Rb (4H1) Mouse mAb(#9303,1:1000). Anti-Nucleolin (phospho T84) antibody (#ab155977,1:1000) and Anti-Rb (phospho T821) antibody (#ab4787,1:1000) were purchased from Abcam. Anti-HaloTag antibody (#G9281,1:1000) and Monoclonal ANTI-FLAG® M2 antibody (#F1804,1:5000) were purchased from Promega and Sigma respectively.

**mRNA expression analysis**. We retrieve the mRNA expression (RNA-seq) dataset (187 samples in 156 cases) of AML and corresponding clinical information from TARGET (Therapeutically Applicable Research to Generate Effective Treatments) using GDC portal. Among them 31 cases have both primary and recurrent samples. We convert the raw read count to transcript per million (TPM) as described before[55]. After that, we retrieved the normalized expression values of CDK2 and GAPDH. The expression values (TPM) of CDK2 and GAPDH in whole blood were retrieved from GTEx[56]. To reduce potential batch effect, we further normalized the expression value of CDK2 relative to the expression value of GAPDH per sample. Wilcoxon Test were used in statistical analysis.

**Real-time PCR**. Total RNA (1 mg), collected by the RNeasy kit (QIAGEN), was used in a reverse transcriptase reaction with the SuperScript III Reverse Transcriptase kit (Life Technologies). The SYBR Green Real-Time PCR Master Mixes kit (Life Technologies) was used for the thermocycling reaction in an ABI-7500 RealTime PCR machine (Applied Biosystems). The real-time PCR analysis was carried out in triplicate with the following primer sets:
CDK2 Forward: 5′-CCAGGAGTTACTTCTATGCCTGA-3′
CDK2 Reverse: 5′-TTCATCCAGGGGAGGTACAAC-3′
DDX5 Forward: 5′-GCTTGCTGAAGATTTCCTGAAAGAC-3′
DDX5 Reverse: 5′-TCTCACTCATGATCTCTTCCATTAGAC-3′
Mcl-1 Forward: 5′-GGACATCAAAAACGAAGACG-3′
Mcl-1 Reverse: 5′-GCAGCTTTCTTGGTTTATGG-3′
STMN1 Forward: 5′-GCCCTCGGTCAAAAGAATCTG-3′
STMN1 Reverse: 5′-TGCTTCAAGACCTCAGCTTCA-3′
GAPDH Forward: 5′-CCACATCGCTCAGACACCAT-3′
GAPDH Reverse: 5′-GCGCCCAATACGACCAAAT-3′

**Nano differential scanning fluorimetry method**. Prometheus NT.48 instrument (NanoTemper Technologies, Munich, Germany) was used to determine Thermal protein unfolding for CDK2 WT and the mutants. The samples were subjected to a temperature gradient of 1 °C/min from 25 °C to 95 °C and the fluorescence was constantly monitored. In all, 10 μl of 0.65 mg/mL sample solution was used for each analysis. Protein unfolding was detected by following the change in fluorescence at emission wavelengths of 330 nm, 350 nm and their ratios. Melting temperature (Tm) was determined using PR. ThermControl v2.0.4 software (NanoTemper Techonlogies) using first derivative analysis of 350 nm/330 nm fluorescence ratio plotted against the temperature.

**Polysome profiling**. Ten million cells were harvested after incubation of 100 μg/mL CHX at 37 °C for 10 min[57]. The cells were washed with ice-cold PBS buffer, and added with 1 mL of Lysis buffer (50 mM Tris, pH 7.5, 150 mM NaCl, 5 mM MgCl2, 1% Triton X-100, 2 mM DTT, 20 U/mL SuperaseIn (Thermo Fisher), 0.5 tablet of Protease inhibitor (Roche), 100 μg/mL CHX), then the lysate was Sheared with a 26-gauge needle gently for 10 times, and centrifuge at 13,000×g for 10 min. The supernatant was loaded on a 15–40% linear sucrose density gradient, and ultracentrifugation at 40,000×g for 2.5 hours in a Beckman SW41 Ti rotor. During ultracentrifugation, the Isco Density Gradient Fractionation System was set up according to the manufacturer's recommendations. Before each experiment, the entire system was washed first with 0.1 M NaOH, then thoroughly with DEPC-treated water. After centrifugation, the centrifuge tube was connected to a flow cell and the piercing apparatus. The 50% sucrose chase solution was injected by puncturing the tube from the bottom, pushing out the gradient in a continuous manner into the flow cell. A typical polysome profile first showed a peak of A254 absorbing material, containing the untranslated mRNAs, then the two peaks of the small and large ribosomal subunits, the monosome peak and finally the polysomal peaks. Polysome profiles can vary according to the general translation activity. All steps were done at 4 °C.

**Survival analysis**. We use R package survival and survminer for survival analysis, median of CDK2 expression level across all the cases involved in TARGET-AML (https://portal.gdc.cancer.gov/projects/TARGET-AML) was set as cutoff (Source data file). For those cases with both primary and recurred samples, the CDK2 expression value in recurred samples were kept in survival analysis.

**Statistical analysis**. Student's $t$ test (two-sided) was applied, and changes were considered statistically significant for $p < 0.05$. For intercomparison of more than 2 groups, a one-way ANOVA followed by a post-hoc test was applied. In the figures, changes are noted using $*p < 0.05$ and $**p < 0.01$. The data were normally distributed and variation within and between groups was not estimated. The sample size was not preselected, and no inclusion/exclusion criteria were used. Survival in mouse experiments was represented with Kaplan–Meier curves, and significance was estimated with the log-rank test (Prism GraphPad). The data shown in the bar graphs are the mean and s.d. of at least three biological replicates. Statistical analysis was conducted using the Microsoft Excel or GraphPad Prism software packages.

**Reporting summary**. Further information on research design is available in the Nature Research Reporting Summary linked to this article.

## Data availability
The authors declare that all data supporting the findings of this study are available within the paper and its Supplementary Information files. The sgRNA sequences targeting human CDK2 were designed using the CRISPR on-line design tool (www.genome-engineering.org/crispr). The mRNA expression and survival results presented in this study are based upon data generated by the Therapeutically Applicable Research to Generate Effective Treatments (https://ocg.cancer.gov/programs/target) initiative,

phs000465. The data used for this analysis are available at https://portal.gdc.cancer.gov/projects. The structure data used in this study are available in the PDB database under accession codes 1FIN, 2IWG, 4Y72. The mass spectrometry proteomics data have been deposited to the ProteomeXchange Consortium via the PRIDE partner repository with the dataset identifier PXD031914. Source data are provided with this paper.

## Code availability

Code for the survival analysis model is available at https://github.com/GuLabZJU/AML. Model code and output data are available from the corresponding author on reasonable request.

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

## Acknowledgements

We thank Dr. Arthur Riggs and Dr. John Chan from City of Hope for their discussion and advice. We thank Dr. Ian Talisman from City of Hope for his help in manuscript editing and proofreading. We thank Dr. He Feng from Zhejiang University for assistance with polysome profiling analysis, and Mass Spec facility from City of Hope for mass spectrometry service. W.H. is supported by the National Cancer Institute under award number NCI 2R01CA139158 and National Institute of Diabetes and Digestive and Kidney Diseases under award number NIDDK R01DK124627. Y.Gu is supported by the National Key R&D Program of China under award number 2018YFA0800100 and 2018YFA0800102, as well as the National Natural Science Foundation of China under award number 31970555. J.Z. is supported by the National Natural Science Foundation of China under award number 32070630. Research reported in this publication was also supported by the National Cancer Institute of the National Institutes of Health under award number P30CA33572.

## Author contributions

W.H. and Y.Gu designed and supervised most of the experiments and wrote the manuscripts. J.Z. and Y.Gan are main figures to perform most of the experiments and data analysis. L.Lin performed the experiments on primary AML samples. J.Y. performed the bioinformatic analysis of the mRNA expression data from public database (TARGET). H.Li developed the computational docking method and performed the screening in silico. X.H. performed the AML mice model experiments. B.K. and Z.F. performed the flow-cytometry experiments. S.X., L.G., L.D., E.Z., and X.M. provided technical assistance for some experiments. Y.X. provided the primary tumor samples. J.L. helped to synthesize HHT-Beads. L.Li, D.H., R.X., and H.Y. provided conceptual and technical advice and support.

## Competing interests

The authors declare no competing interests.
