## [Peer Review File · Nature Communications]

Inhibition of the CDK2 and Cyclin A complex leads to autophagic degradation of CDK2 in cancer cellsReviewers' comments:

Reviewer #1 (Remarks to the Author): Expert in Cdk inhibitors

This paper by Zhang et al. proposes a novel mechanism of action of HHT. They provide evidence that HHT binds directly to CDK2 and that this in turn prevents cyclin binding and leads to CDK2 degradation. The conclusions are original and there are some interesting new findings regarding HHT and CDK2.

These include:

- a potential binding mode for HHT on CDK2
- Decreased CDK2 levels by HHT in various cancer cells that may occur via the autophagy-lysosome pathway
- an interaction between CDK2 and Trim21

The manuscript also has several serious shortcomings.

It seems likely based on the data that HHT can interact with CDK2, but the evidence that this event is critical for some of the various effects of HHT on cells shown here, much less the clinical activity of HHT, is not very convincing. Furthermore, in its current form the evidence is hard to properly interpret because key details are very frequently absent. Most glaring, units for concentration of HHT are often not provided and in many cases no value is given at all. With the exception of Fig 3A, where HHT concentrations are reported as ng/mL, I could not find concentration units explicitly mentioned anywhere else save one other reference to 'nM'. The reader is left having to assume that perhaps all other figures with concentration numbers are ng/mL but wondering whether they may in fact be nM or something else. Micromolar? In many of the time courses we don't even find a number so have absolutely no idea what concentration of HHT was used. Maybe I missed it, but I couldn't find this information in the text, methods, figures, or figure legends. Ideally it should be in at least two of those. To understand whether the results might have clinical relevance, or whether they might be useful as starting points for novel approaches to target CDK2, the concentration required for the effects is a critical detail.

Unfortunately, and as the authors point out, the mutant CDK2 (CDK2-3As) was not active (could not rescue the CDK2 knockout) making it far from ideal as a reagent to determine whether HHT-CDK2 binding is important for the effects of HHT in cells. So, for example, in the cell survival data on Fig 2, CDK2-3As is not useful as a rescue reagent and so the conclusion from this experiment that 'HHT may functionally inhibit CDK2, which in turn may inhibit cancer cell proliferation' is not strongly supported by these data. Fig 2F is consistent with the conclusion but here a problem is that the CDK2^{-/-} cells grow slower. Were the viability data normalized for growth rate? If not, that could explain the different dose response to HHT. Also, I had to presume that "THP1 CDK2" and "THP1 CDK2-3As" cells are actually THP1(CDK2^{-/-}) cells expressing the CDK2 and CDK2-3As constructs rather than simply THP1 parent cells with those constructs. The legend and text didn't make that clear to me.

The statement that "HHT strongly decreased the half-life of HT-CDK2 fusion protein" just does not appear justified based on the data in Fig 3C and S5.

In Fig 4F is the '+' and '++' concentration of HHT listed the same in each sample (again, it would be good to know exactly what the concentration is)?

IC50s are described on lines 168 and 169 and Fig 4G but nowhere could I find reference to how those IC50s were measured. What assay?

Reviewer #2 (Remarks to the Author): Expert in autophagy

This is a potentially interesting study that identifies a unique mechanism of action for homoharringtonine (HHT), a drug that has demonstrated some utility in the treatment of leukemias. The possibility that this agent induces the degradation of the CDK2 protein could open the possibility for its uses in other malignancies where activation of CDK2 mediated signaling pathways is thought to be responsible for evasion of growth arrest by conventional or targeted therapies. Unfortunately, there are a number of concerns which markedly reduce enthusiasm for this manuscript. The primary reservation relates to the concentration(s) of HHT that are used in the different studies. It would be obligatory to limit the concentration used to that which can be achieved in the clinic, which is approximately 40ng/ml with multiple doses and 25ng/ml with a single dose (Nemunaitis et al). Consequently, the dose responses presented in Figure 2B in PC3 and HeLa cells using concentrations that are 10 fold to 25 fold higher are essentially irrelevant and suggest that this agent is unlikely to be appropriate for many solid tumors. This same concern applies to e.g. the data presented in Figure 3A in the HeLa and PC3 cell lines and in Figures 7A, 7B and 7C where the concentrations used are (presumably) 300 ng/ml and 600 ng/ml. In other studies, such as for example, Figure 6, HHT concentrations are not indicated in the figure or figure legend. Overall, the use of different drug concentrations throughout the manuscript makes it virtually impossible to compare and comprehend data in different experiments.

Another fundamental concern is the reliance on the MTS assay as an indication of drug sensitivity. This assay does not directly distinguish between growth arrest (cytostatic drug effects) and cell death (cytotoxic drug effects). One cannot discern from the data in Figures 2G whether the HHT is only producing growth arrest (which might be expected if the only effect is at the level of cdk2) or cell death, which might be occurring via the multiple other targets that have been established for this agent in previous literature.

Figure 4 evaluates the impact of HHT on tumor development. However, the critical ((translational) questions is the impact on an established tumor, which has not been determined.

How the data were generated for Figure 4D is not explained. What cell types are indicated by the different data points?

The findings relating to drug effects on autophagy signaling are intriguing. However, the studies in e.g. Figure 7, fail to link the proposed mechanistic elements with regulation of autophagy since no experiments were performed in cells where autophagy was genetically compromised. However, and this cannot be emphasized forcefully enough, these experiments are likely to be meaningless if the studies were actually performed at HHT concentrations of 300ng/ml and 600ng/ml.

Point-by-point response to the reviewers' comments

Reviewer #1 (Remarks to the Author): Expert in Cdk inhibitors

1. This paper by Zhang et al. proposes a novel mechanism of action of HHT. They provide evidence that HHT binds directly to CDK2 and that this in turn prevents cyclin binding and leads to CDK2 degradation. The conclusions are original and there are some interesting new findings regarding HHT and CDK2. These include:

- *a potential binding mode for HHT on CDK2*
- *Decreased CDK2 levels by HHT in various cancer cells that may occur via the autophagy-lysosome pathway*
- *an interaction between CDK2 and Trim21*

Reply: Thanks very much for the positive comments.

2. It seems likely based on the data that HHT can interact with CDK2, but the evidence that this event is critical for some of the various effects of HHT on cells shown here, much less the clinical activity of HHT, is not very convincing. Furthermore, in its current form the evidence is hard to properly interpret because key details are very frequently absent.

Reply: The reviewer's comment is very critical. Our study has identified HHT as a disruptor of CDK2 complex to promote the degradation of CDK2. As suggested, we have added the required detailed information. To test whether this novel mechanism is important for the anti-leukemia activities of HHT, we performed additional experiments as shown below.

In new Fig.2, we did the tumor xenograft experiment. Briefly, the same number of THP-1 cells and THP-1^{CDK2-/-} cells with stable expression of luciferase were inoculated to NOD/SCID mice. After one week, the mice started to receive the treatment of HHT or vehicle control. The development of leukemia was then monitored by luminescence detection. The results showed that THP-1 tumors were growing much faster than THP-1^{CDK2-/-} tumors. As expected, THP-1 tumors showed more sensitive to HHT treatment compared to THP-1^{CDK2-/-} tumors, which suggested that CDK2 was critical for both leukemia development and anti-leukemia effects of HHT *in vivo* (Fig. 2G&H).

Fig. 2. (G) Bioluminescence imaging of luciferase-expressing THP1 (CDK2 wild-type or knockout) tumor xenograft models. The mice were treated with 0.5 mg/kg HHT (n=6). (H) Kaplan–Meier survival of tumor xenograft models inoculated with luciferase-expressing THP1 cells (CDK2 wild-type or knockout). The mice were treated with 0.5 mg/kg HHT or vehicle as the control. * $p < 0.05$, ** $p < 0.01$, *** $p < 0.001$.

Moreover, we measured the expression level of CDK2 based on RNA-seq datasets and found that CDK2 mRNA expression level increased significantly in AML cohort in comparison with whole blood samples in normal cohort ($p < 0.0001$) (Fig 2A). Meanwhile, we also observed that the mRNA expression level of CDK2 was also significantly increased in recurred samples relative to the corresponding primary samples ($n = 31$, $p < 0.05$) (Fig 2B). We then further evaluated the elevated expression level of CDK2 with overall survival in AML patients. As expected, AML patients ($n = 156$, $p = 0.007$) that express higher CDK2 show significantly poorer overall survival (Fig 4H). This analysis indicates that CDK2 may play a critical role in leukemia development.

Fig. 2. (A) Normalized expression level (TPM) of CDK2 relative to GAPDH in TARGET-AML and GTEx cohort. (B) Normalized expression level (TPM) of CDK2 relative to GAPDH in primary and recurred samples using TARGET-AML (n = 31).

Fig. 4. (H) The overall survival of CDK2 for human AML patients with high (red, n = 78) versus low (blue, n = 78). The patients were dichotomized on the basis of the median value of CDK2 mRNA expression. Wilcoxon Test was used to estimate the difference of expression level, and Log Rank test was used for survival analysis.

3. Most glaring, units for concentration of HHT are often not provided and in many cases no value is given at all. With the exception of Fig 3A, where HHT concentrations are reported as ng/mL, I could not find concentration units explicitly mentioned anywhere else save one other reference to 'nM'. The reader is left having to assume that perhaps all other figures with concentration numbers are ng/mL but wondering whether they may in fact be nM or something else. Micromolar? In many of the time courses we don't even find a number so have absolutely no idea what concentration of HHT was used. Maybe I missed it, but I couldn't find this information in the text, methods, figures, or figure legends. Ideally it should be in at least two of those. To understand whether the results might have clinical relevance, or whether they might be useful as starting points for novel approaches to target CDK2, the concentration required for the effects is a critical detail.

Reply: We apologize for the previously unclear description. In the revised manuscript, we have added the concentrations of HHT in each experiment using the same unit (ng/ml).

4. Unfortunately, and as the authors point out, the mutant CDK2 (CDK2-3As) was not active (could not rescue the CDK2 knockout) making it far from ideal as a reagent to determine whether HHT-CDK2 binding is important for the effects of HHT in cells. So, for example, in the cell survival data on Fig 2, CDK2-3As is not useful as a rescue reagent and so the conclusion from this experiment that 'HHT may functionally inhibit CDK2, which in turn may inhibit cancer cell proliferation' is not strongly supported by these data. Fig 2F is consistent with the conclusion but here a problem is that the CDK2^{-/-} cells grow slower. Were the viability data normalized for growth rate? If not, that could explain the different dose response to HHT.

Reply: Thanks for the critical comments. We agree that CDK2-3As mutant is not an ideal reagent to directly demonstrate that HHT-CDK2 binding is important for the effects of HHT on cells, due to its lack of kinase activity. We thus created a kinase active mutation at 160 (T160E) in CDK2-3As (named as CDK2-3As/T160E hereafter) to mimic the phosphorylation of Thr160, which is located in the activation loop of CDK2 and its phosphorylation is critical for the kinase activity. Compared to the wild-type CDK2, the CDK2-3As/T160E showed comparable kinase activity on Nucleolin (NCL) (Fig. S5D).

Fig. S5. (D) Representative western blot of CDK2 downstream target protein-NCL in THP1 wild-type CDK2, CDK2-3As and CDK2-3As/T160E cells.

The proliferation of THP1 CDK2-3As/T160E cells was comparable to that of THP1 wild-type CDK2 cells during the first four days, and after that wild-type CDK2 cells exhibited rapid proliferation from day 5 (Fig 2E). As expected, the THP1 CDK2-3As/T160E cells showed highly resistant to HHT treatment compared to wild-type CDK2 cells (Fig. 2F). These experiments suggested that the HHT-CDK2 binding is important for the effects of HHT in THP1 cells.

Fig. 2. **(E)** Growth curves of THP1 CDK2 and THP1 CDK2-3As/T160E cells. **(F)** Cell survival curve of THP1 CDK2 and THP1 CDK2-3As/T160E cells after HHT treatment for 24 hours.

More importantly, our data have shown that the interaction between CDK2 and HHT is critical for the anti-leukemia effects of HHT *in vivo* (Fig. 2G&H). Thus, with these results the CDK2-3As mutant could at least indirectly demonstrate the importance of HHT-CDK2 binding for the effects of HHT on cells.

Fig. 2. **(G)** Bioluminescence imaging of luciferase-expressing THP1 (CDK2 wild-type or knockout) tumor xenograft models. The mice were treated with 0.5 mg/kg HHT (n=6). **(H)** Kaplan–Meier survival of tumor xenograft models inoculated with luciferase-expressing THP1 cells (CDK2 wild-type or knockout). The mice were treated with 0.5 mg/kg HHT or vehicle as the control. * p < 0.05, ** p < 0.01, *** p < 0.001.

We confirm that the viability data were not normalized for growth rate in Fig. 2F&G (previous version) and Fig. 2D&F (current version).

5. Also, I had to presume that "THP1 CDK2" and "THP1 CDK2-3As" cells are actually THP1(CDK2^{-/-}) cells expressing the CDK2 and CDK2-3As constructs rather

than simply THP1 parent cells with those constructs. The legend and text didn't make that clear to me.

Reply: We apologize for the elusive description. In current version, we have made clear description in both the figure legends and text. We used a CRISPR-Cas9 approach to delete CDK2 in THP1 cell line (THP1^{CDK2-/-}). We then introduced either exogenous wild type (WT) CDK2, mutant CDK2-3As or mutant CDK2-3As/T160E in the THP1^{CDK2-/-} cells by lentivirus infection, naming them as “THP1 CDK2” cells, “THP1 CDK2-3As” cells and “THP1 CDK2-3As/T160E” cells, respectively.

6. The statement that "HHT strongly decreased the half-life of HT-CDK2 fusion protein" just does not appear justified based on the data in Fig. 3C and S5.

Reply: We apologize that we may not have described the HaloTag (HT) Pulse-Chase Assay very clear. The HaloTag is a tag protein which could covalently bind to a specific fluorescent ligand (Promega). After 15-minute incubation with this ligand, all the synthesized HT-CDK2 protein binds to this ligand and have the fluorescence, but not the new synthesized protein. Therefore, we can determine the fluorescence intensity to monitor the protein stability. To make it easier to read, we put the two figures together in current version (Fig. 3C&D), and the results showed that fluorescence intensity decreased in HT-CDK2 significantly during HHT treatment, but not in the HT-CDK2-3As, suggesting that HHT could decreased the half-life of HT-CDK2 fusion protein.

Fig. 3. (C) Pulse-chase analysis of exogenous CDK2 protein using HT-TMR system in 293T cells transfected with HT-CDK2. The HT-TMR ligand-labeled HT-CDK2 was visualized with a fluoro-image analyzer (upper panel) and the total expression of HT-CDK2 was analyzed by western blot with GAPDH as the loading control. The relative amount of HT-TMR ligand-labeled HT-CDK2 was measured with densitometric intensity of each band (lower panel). (D) Pulse-chase analysis of exogenous CDK2-3As protein using HT-TMR system in 293T cells transfected with HT-CDK2-3As. The HT-TMR ligand-labeled HT-CDK2-3As was visualized with a fluoro-image analyzer (upper panel) and the total expression of HT-CDK2-3As was analyzed by western blot with GAPDH as the loading control. The relative amount of HT-TMR ligand-labeled HT-CDK2-3As was measured with densitometric intensity of each band (lower panel).

7. In Fig 4F is the '+' and '++' concentration of HHT listed the same in each sample (again, it would be good to know exactly what the concentration is)?

Reply: As suggested, we have now added the exact concentrations of HHT in the Fig. 4F.

Fig. 4. (F) Western blot analysis of CDK2 expression in BMCs of indicated human primary AML specimens (E) treated with or without HHT as indicated.

8. IC50s are described on lines 168 and 169 and Fig. 4G but nowhere could I find reference to how those IC50s were measured. What assay?

Reply: We apologize for the confusion. We used the MTS assay to measure the cytotoxic effects of HHT on primary leukemia cells at different doses, and the IC50 value was calculated by GraphPad Prism software. We have added the information in the revised manuscript.

Reviewer #2 (Remarks to the Author): Expert in autophagy

1. This is a potentially interesting study that identifies a unique mechanism of action for homoharringtonine (HHT), a drug that has demonstrated some utility in the treatment of leukemias. The possibility that this agent induces the degradation of the CDK2 protein could open the possibility for its uses in other malignancies where activation of CDK2 mediated signaling pathways is thought to be responsible for evasion of growth arrest by conventional or targeted therapies.

Reply: Thanks very much for the positive comments.

2. Unfortunately, there are a number of concerns which markedly reduce enthusiasm for this manuscript. The primary reservation relates to the concentration(s) of HHT that are used in the different studies. It would be obligatory to limit the concentration

used to that which can be achieved in the clinic, which is approximately 40ng/ml with multiple doses and 25ng/ml with a single dose (Nemunaitis et al). Consequently, the dose responses presented in Figure 2B in PC3 and HeLa cells using concentrations that are 10 fold to 25 fold higher are essentially irrelevant and suggest that this agent is unlikely to be appropriate for many solid tumors. This same concern applies to e.g. the data presented in Figure 3A in the HeLa and PC3 cell lines and in Figures 7A, 7B and 7C where the concentrations used are (presumably) 300 ng/ml and 600 ng/ml. In other studies, such as for example, Figure 6, HHT concentrations are not indicated in the figure or figure legend. Overall, the use of different drug concentrations throughout the manuscript makes it virtually impossible to compare and comprehend data in different experiments.

Reply: Regarding the concentrations of HHT used in the experiments,

- (1) We revisited our previous data and found that previously established THP1 CDK2 and THP1 CDK2-3As cells had greater expression of CDK2 protein compared to the wild-type THP1 cells, which made the cells more resistant to HHT treatment as shown below. Therefore, we did the stable overexpression again with lower MOI, and the new established overexpressed cell lines showed comparable CDK2 expression level with wild-type THP1 cells (Fig. S5A). We then repeated the experiments with clinically achievable doses of HHT, and the new results are similar to the previous results.

Figure not shown in manuscript: Representative western blot of CDK2 protein in THP1 cells, THP1^{CDK2-/-} cells and THP1^{CDK2-/-} cells with re-expression of CDK2 or CDK2-3As, which were established previously.

Figure S5. (A) Representative western blot of CDK2 protein in THP1 cells, THP1^{CDK2-/-} cells and THP1^{CDK2-/-} cells with re-expression of CDK2 or CDK2-3As, which were established currently.

- (2) PC3 and HeLa cells did demonstrate more resistance to HHT treatment compared to THP-1 cells. But we found that lung cancer cell line A549 and Hepatocarcinoma cell line Hep3B were very sensitive to HHT treatment as shown in Figure S3. And the response difference between the two groups of cell lines may reflect the differential expression level of CDK2 and Trim21 as shown below: HeLa and PC3 cells have higher CDK2 protein level and lower Trim21 protein level compared to A549 and Hep3B cells.

This suggests that we need to dissect the expression level of CDK2 and Trim21 before we apply HHT on solid tumor cells.

Fig. S3. Cell survival curve based on MTS assay for A549 and Hep3B cell lines treated with HHT for 24 hours.

Figure not shown in manuscript: Representative western blot of CDK2 and Trim21 proteins in different cancer cell lines.

(3) As pointed out by the reviewer #1, the HHT concentrations are not described clearly in the figures or figure legends. We have now made it clear in the revised manuscript.

3. Another fundamental concern is the reliance on the MTS assay as an indication of drug sensitivity. This assay does not directly distinguish between growth arrest (cytostatic drug effects) and cell death (cytotoxic drug effects). One cannot discern from the data in Figures and 2G whether the HHT is only producing growth arrest (which might be expected if the only effect is at the level of *cdk2*) or cell death, which might be occurring via the multiple other targets that have been established for this agent in previous literature.

Reply: Thanks for the insightful comments. Cells with different CDK2 status were treated with HHT for 24 hours at 40 ng/mL and then subjected to Annexin-V/DAPI staining assay to determine the cell apoptosis. The results showed that both cells have increased Annexin V positive population after 24 hours treatment compared to negative controls, but the increased apoptosis was not enough to explain the effect of HHT on leukemia cell proliferation, and there was no difference between THP1 and THP1^{CDK2-/-} cells. These results indicate that growth arrest by inhibiting CDK2 may be a substantial effect of HHT on leukemia cells.

Figure not shown in manuscript: Representative result of Annexin-V/DAPI staining assay to detect the apoptotic rate in THP1 and THP1^{CDK2}^{-/-} cells with HHT treatment for 24 hours and the negative control with DMSO treatment.

4. Figure 4 evaluates the impact of HHT on tumor development. However, the critical ((translational) questions is the impact on an established tumor, which has not been determined.

Reply: In new Fig.2, we did the tumor xenograft experiment. The same number of THP-1 cells and THP-1^{CDK2}^{-/-} cells with stable expression of luciferase were inoculated to NOD/SCID mice. After one week, the mice started to receive the treatment of HHT or vehicle control. The development of leukemia was then monitored by luminescence detection. The results showed that THP-1 tumors were growing much faster than THP-1^{CDK2}^{-/-} tumors. As expected, THP-1 tumors showed more sensitive to HHT treatment compared to THP-1^{CDK2}^{-/-} tumors did, which suggested that CDK2 was critical for both leukemia development and anti-leukemia effects of HHT in vivo (Fig. 2G&H).

Fig. 2. (G) Bioluminescence imaging of luciferase-expressing THP1 (CDK2 wild-type or knockout) tumor xenograft models. The mice were treated with 0.5 mg/kg HHT (n=6). (H) Kaplan–Meier survival of tumor xenograft models inoculated with luciferase-expressing THP1 cells (CDK2 wild-type or knockout). The mice were treated with 0.5 mg/kg HHT or vehicle as the control. * $p < 0.05$, ** $p < 0.01$, *** $p < 0.001$.

5. How the data were generated for Figure 4D is not explained. What cell types are indicated by the different data points?

Reply: We apologize for the unclear description. The experiment in Figure 4D determined the CDK2 protein level in bone marrow cells collected from MLL-AF9-induced leukemia mice treated with either HHT or vehicle for 10 weeks. We have added the new information in the text and figure legend.

6. The findings relating to drug effects on autophagy signaling are intriguing. However, the studies in e.g. Figure 7, fail to link the proposed mechanistic elements with regulation of autophagy since no experiments were performed in cells where autophagy was genetically compromised. However, and this cannot be emphasized forcefully enough, these experiments are likely to be meaningless if the studies were actually performed at HHT concentrations of 300ng/ml and 600ng/ml.

Reply: Thanks very much for the suggestion. We have done the ATG7 knockdown assay by siRNA, followed by HHT treatment with reasonable concentrations. The results showed that when the autophagy was suppressed by ATG7 knockdown, the

CDK2 protein level was rescued during HHT treatment, which indicated that HHT induced CDK2 protein degradation was autophagy dependent.

Fig. 5. (D) THP-1 cells were transfected with siRNA targeting on ATG7 or GFP as the negative control. Two days after siRNA transfection, the cells were treated with HHT for 9 hours and the protein level of CDK2 was analyzed by western blot. GAPDH was the loading control.

Thanks for the comments on HHT concentrations. As we have replied previously in point #2.1, We revisited our previous data and found that previous established THP1 CDK2 and THP1 CDK2-3As cells had greater expression of CDK2 protein compared to the wild-type THP1 cells, which made the cells more resistant to HHT treatment as shown below. Therefore, we did the stable overexpression again with lower MOI, and the new established overexpressed cell lines showed comparable CDK2 expression level with wild-type THP1 cells (Fig. S5A). We then repeated the experiments with clinically achievable doses of HHT, and the new results are similar to the previous results.

Figure not shown in manuscript: Representative western blot of CDK2 protein in THP1 cells, THP1^{CDK2-/-} cells and THP1^{CDK2-/-} cells with re-expression of CDK2 or CDK2-3As, which were established previously.

Figure S5. (A) Representative western blot of CDK2 protein in THP1 cells, THP1^{CDK2-/-} cells and THP1^{CDK2-/-} cells with re-expression of CDK2 or CDK2-3As, which were established currently.

REVIEWER COMMENTS

Reviewer #2 (Remarks to the Author):

The authors appear to have made a conscientious effort to address the concerns raised in the review. For instance, the additional experimental data relating to genetic autophagy inhibition provides critically relevant proof of concept. However, there is a persistent lack of clarity involving the issue of drug concentration used.

In the previous review, it was noted that some of the figures appeared to utilize inappropriately high drug concentrations. Were these figures removed? The absence of a marked version of the manuscript indicating modifications makes it difficult to discern what changes were made.

In some of the current figures such as in the Supplementary data, drug concentrations are still not consistently indicated.

The authors need to clearly indicate what drug concentrations were used in each experiment, and where the drug concentrations are clinically relevant. This issue needs to be addressed and discussed in the text as well.

Figures that were provided in the response to the previous review should be included in the manuscript with an explanation of why these studies were necessary and their implications. These should not be solely for the benefit of the reviewer. Just one example is the data on apoptosis that is not included in the revised manuscript.

Reviewer #4 (Remarks to the Author):

Review of Autophagic Degradation of CDK2 Protein in Cancer Cells for Nature Communications.

In their manuscript, J. Zhang et al. search for new, non-ATP competitive chemical inhibitors of CDK2. Using an in silico screen, they identify homoharringtonine (HHT) as a candidate drug. The binding of HHT to CDK2, as well as the disruption of CDK2 binding to cyclin A & E, is validated by several in vitro assays. The specificity of HHT towards CDK2 and the efficacy of HHT in inhibiting cancer cell growth is validated using several genetically modified versions of CDK2 in cell culture assays, as well as by one in vivo assay. HHT's mechanisms of action if then studied in cultured cells, where the authors find evidence for HHT treatment resulting in CDK2 degradation. This degradation of CDK2 is shown to be autophagy-dependent and at least partially mediated by Trim21.

Overall, the manuscript describes new and interesting results, which could promote the study of CDK2 functions, suggest a new degradation mechanism for CDK2 and possibly even promote the development of new pharmaceuticals. If properly validated and carefully presented, these discoveries could warrant publication in Nature communications. Some experiments in the manuscript are very good, including the generation of CDK2-3As/T160E cell line to validate that HHT seems to act, at least partly, through CDK2.

However, in its current form the manuscript's work is completely unreproducible and a few key biological considerations have been ignored. Below, I have listed key issues, which should be addressed before publication in any high-quality journal could be considered. Notably, many of these can be addressed by writing. I have also listed more minor issues, which would help improve the quality and impact of the work, although they are less critical for accepting this work for publication.

Key issues for publication:

- Homoharringtonine, similarly to its homolog harringtonine, is a translation inhibitor that block the elongation phase or ribosomes. The authors did not even mention this in the manuscript, despite the fact that this may explain many of the results authors observe. In cell-free assays even low nanomolar concentrations of HHT can inhibit translation (Selective inhibition of the polypeptide chain elongation in eukaryotic cells. 1992. Tujebajeva et al. PMID: 1730056), so it is critical for the authors to both discuss this in the manuscript and test if their discoveries are independent of any effects on protein synthesis. This could be done, for example, by comparing HHT's protein synthesis inhibition potential to HHT's potential in disrupting CDK2's binding to cyclin and to HHT's potential in stimulating CDK2 degradation. These experiments should be done carefully with several incubation times and model systems, including the CDK2-3As/T160E cell line in order to validate that protein synthesis inhibition is not due to HHT's effects on CDK2. If protein synthesis is not affected (or only very minimally affect) by the concentrations of HHT used in the study across multiple model systems, then the manuscript's findings are very interesting.

- The Methods section is severely lacking to a point that the study could not be reproduced at all. Without these corrections I would not recommend this manuscript for publication in any journal. Below is a list of missing details. Note that this list may not be comprehensive and the authors should make a serious effort in verifying that relevant details are presented.

- o Many chemical concentrations used, as well as some treatment times are still not reported (MG132, Roscovitine, NH₄Cl, etc.).
- o Antibodies used and labeling details are not listed at all.
- o LIVS method details are very superficial. More importantly, it is unclear if this is a new method, in which case the method section should be significantly more detailed, or if this is a previously established method, in which case references are missing.
- o Pulldown experiment details are not presented.
- o It is unclear where patient data was obtained from.
- o siRNA details and usage information is completely missing.

- Reproducibility of most results is unclear. How many times were each experiment repeated? Optimally, the authors would show quantifications of all replicates for western blots next to the actual blots, especially for key figures.

- Experiments in figure 2G and 2H do not proof that HHT has to act through CDK2 in vivo, because the CDK2 KO model had almost influence on animal survival in the first place. Thus, it remains unclear if HHT would have also increased survival in the CDK2 KO model, especially as the experiment relied on very few mice. Optimally, this would have been done using the CDK2-3As/T160E model, with longer-term examination of the mice and with larger replicate numbers.

Minor points to improve the work:

- The manuscript could use proof reading and careful validation of references. For example, the abstract misspells 'cyclin' as 'cylcin', and when discussing "aberrant activation of CDK2" (line 49 & 50 in intro) the authors reference ref#9 which has no data on any 'aberrant activation of CDK2'.
- Some of the claims in the manuscript need to be dialed back, especially for non-specialist audiences. In silico predictions of drug binding should be discussed as predictions until comprehensive experimental validations are carried out.
- The authors validated that CDK1 does not seem to be affected by HHT. However, the CDK2 phosphorylation targets studied are also phosphorylated by the closely related CDK4. The results could be influenced by HHT inhibiting CDK4. This should be preferably tested or at least acknowledged in writing.
- Regarding data in figure 4 A-D, the authors state that "the ability of HHT to effectively inhibit the development of leukemia in a mouse model appeared to involve the degradation of CDK2 protein." (line 208). Technically, no causality between development of leukemia and CDK2 degradation is established in these experiments and the conclusion is therefore an overstatement.
- Experiments in figure 5C. MG132 and NH4Cl are recognized as somewhat dirty drugs with many off-targets. These experiments would be more convincing if the authors repeated them with other proteasome and autophagy inhibitors.
- On lines 253 and 254 the authors discuss the discoveries of their MS/MS experiment. Where is the data?
- Figure S9B requires quantifications and controls (wt cells) should also be shown next to the two mutants.
- The microscopy in the last main figure is extremely limited in its usefulness, as no replicates are shown, no methods are listed and no quantifications are done. Essentially, the manuscript gains nothing from these images, although more careful investigation could be interesting. Otherwise, even removal of these images could be a better option.
- The discussion claims that "When CDK2 inhibitors are added to a range of established regimens, the inhibitors not only improve the responses of some cancer patients but also overcome other patients' resistance to therapy." (line 322) This requires references. Furthermore, this is soon followed by statement that "The vast majority of CDK2 inhibitors target the ATP binding site of CDK2. ...These inhibitors failed to meet expectations during clinical trials because they have either poor specificity or high toxicity." This seems very contradictory to the previous sentence, especially in the absence of references.
- The discussion states that "While other computational models typically produce a hit rate of approximately 3%, the hit rate for LIVS is nearly 15%." (line 341). This is not clear in the results section and it seems that the authors are discussing the number of predictions of made by LIVS. Yet, having more predictions is not useful if they are incorrect, as seems to be the case for most predictions in this study, as shown in figure S1. Thus, this sentence seems very misleading, unless the authors specify that they are discussing the 'non-validated hit rate'.
- The discussion is largely a listing of the results of the manuscript. This seems like a missed opportunity for the authors to discuss the larger scale implications of their work. Could HHT be easily modified to increase its specificity to CDK2? Which other diseases might benefit from CDK2 inhibition? Does the

autophagic degradation of CDK2 suggest that other CDKs are likely to be degraded by autophagy too?
Etc...

Best wishes,
Reviewer: Teemu P. Miettinen

Reviewer #5 (Remarks to the Author):

Zhang et al. propose that HHT, a known therapeutical, is interacting with CDK2 to disrupt Cyclin A binding and thus inactivates CDK2. Moreover, they propose that binding of HHT to CDK2 induces its degradation via an autophagy lysosomal pathway. They could show in addition that Trim21, a protein involved in the aforementioned pathway interacts with CDK2 in a way that is mutually exclusive with CyclinA binding. Overall, this a well conducted study that opens interesting new possibilities in the repurposing of HHT.

However, in addition to the previous reviewers comments some issues need to be resolved prior to publication.

Although, the methodology how the authors identified HHT as a possible CDK2 binding compound is sound and seems plausible the follow up characterization of the mechanism of action still lacks some experimental conformation.

The authors propose that HHT binds to a certain area on the surface with a predicted hydrogen bonding pattern. To interrupt binding they generated a variant that contains alanine exchanges for residues T47, R50 and R150. In pulldown experiments they show that this variant cannot bind to HHT beads anymore suggesting a disrupted interaction. It is, however, unclear whether this lack of interaction occurs due to the disruption of binding to HHT via the proposed interface or whether this variant is properly folded. A closer look at the structure suggests that this triple variant might significantly influence the local fold. The authors should thus perform comparative folding studies to ensure structural integrity of the variant. This could be accomplished for example either by CD spectroscopy or Thermofluor studies. This analysis should be also accompanied by in vitro kinase/ATPase assays with the recombinant protein in the presence and absence of Cyclin A and HHT. In addition to demonstrating that the fold is maintained, the kinase/ATPase analysis would also yield direct evidence of the mechanism of action which would strengthen their point that HHT disrupts the Cyclin A interface. The results represented in the manuscript are a combination of HHT resin association of CDK2 and pull-down experiments that have been performed in cell based experiments. They show a clear tendency towards the proposed mechanism of action concerning HHT but do not unambiguously prove it. The IC50 values of the suggested experiments would also put the in vivo IC50 values into context that have been obtained with the MTS assay.

With respect to the T160E variant of the CDK2-3As variant it is entirely unclear how this variant fully reactivates CDK2 activity. Aren't both T-loop phosphorylation and cyclin binding both equally important to reach full activity for CDKs? How this variant is able to compensate the previous phenotype of the

CDK2-3As is elusive at this point and should be explained more thoroughly, accompanied by the already proposed in vitro kinase/ATPase studies.

Another concern are the experiments concerning the CDK2 Δ 171-243 variant. It is most likely that the resulting protein is not suitable for any meaningful interpretation since a significant portion of the C-terminal kinase lobe is missing which almost certainly disrupts the folding of CDK2. These results should therefore be interpreted with extreme caution. In addition, the authors propose that the region V154/P155/M233/P234/P238 could form hydrogen-bonds to Trim24, which is most likely a typo and should be corrected. To investigate this in detail a 5R variant (V154R/P155R/M233R/P234R/P238R) of this particular region was constructed to disrupt this interaction. Since none of these residues can undergo side chain hydrogen bonding interactions it must be assumed that the authors only observed main chain interactions with CDK2 at these positions? These are notoriously difficult to tackle and hence they have chosen a drastic approach by changing all residues to an arginine. The three prolines in this region could be of structural importance and again mutations could severely affect the fold in this region rendering their results less interpretable. Here again the analysis of proper folding as recommended for the other variants seems to be essential.

Minor comment:

It would be helpful if the authors could indicate the HHT binding region in Figure 5J of the model, since it should not interfere with the Trim21 binding region.

Point-to-point response to reviewers' comments

Reviewer #2 (Remarks to the Author):

1. The authors appear to have made a conscientious effort to address the concerns raised in the review. For instance, the additional experimental data relating to genetic autophagy inhibition provides critically relevant proof of concept.

However, there is a persistent lack of clarity involving the issue of drug concentration used:

- 1) In the previous review, it was noted that some of the figures appeared to utilize inappropriately high drug concentrations. Were these figures removed? The absence of a marked version of the manuscript indicating modifications makes it difficult to discern what changes were made.
- 2) In some of the current figures such as in the Supplementary data, drug concentrations are still not consistently indicated.
- 3) The authors need to clearly indicate what drug concentrations were used in each experiment, and where the drug concentrations are clinically relevant. This issue needs to be addressed and discussed in the text as well.

Response: We apologize for the missing details of the drug concentrations. We have now added the drug concentration and incubation time used in each experiment, which is highlighted in the revised manuscript.

In the first version, we have applied the HHT treatments to some solid tumor cells, and the results demonstrated antitumor effects at higher dose of HHT than that could be achieved clinically. This may be partially due to the differential expression of CDK2 and Trim21 in some solid tumor cells, which needs further investigation in the future. Therefore, in addition to previous results on HeLa and PC3 cells, we added in the supplementary data the results on A549 (lung cancer cell line) and Hep3B (hepatocarcinoma cell line), demonstrating their sensitivity to clinically achievable dose of HHT.

Fig. S3. Cell survival curve from MTS assay for different cell lines treated with HHT for 24 hours. The results shown here were representative of three independent experiments.

As suggested, we have addressed and discussed in the revised manuscript that the concentrations of HHT used in our experiments are achievable in the clinic, which is approximately 40ng/ml with multiple doses and 25ng/ml with a single dose, as described: "The concentration of HHT applied here could be achieved in the clinic, which is approximately 40ng/ml with multiple doses and 25ng/ml with a single dose²¹."

Ref^{#21}: Nemunaitis J, et al. Pharmacokinetic study of omacetaxine mepesuccinate administered subcutaneously to patients with advanced solid and hematologic tumors. *Cancer chemotherapy and pharmacology* 71, 35-41 (2013).

As suggested, all changes regarding the HHT treatments in this revised manuscript are highlighted.

2. Figures that were provided in the response to the previous review should be included in the manuscript with an explanation of why these studies were necessary and their implications. These should not be solely for the benefit of the reviewer. Just one example is the data on apoptosis that is not included in the revised manuscript.

Response: Thanks for the insightful comments. We have 3 figures in the previous point-to-point response letter that were not included in the manuscript. The first figure as shown below was replaced with Fig. S5A, therefore we did not add it into the revised manuscript to avoid

repetition.

Figure not shown in manuscript: Representative western blot of CDK2 protein in THP1 cells, THP1^{CDK2-/-} cells and THP1^{CDK2-/-} cells with re-expression of CDK2 or CDK2-3As, which were established previously.

The second figure as shown below demonstrated the differential protein levels of CDK2 and Trim21 in different cancer cell lines. This result showed that HeLa and PC3 cells with higher CDK2 protein level and lower Trim21 protein level were more resistant to HHT treatment compared to A549 and Hep3B cells, indicating the possible correlation between sensitivity to HHT treatment and the expression of CDK2 and Trim21. However, it is preliminary and need for further study in the future, therefore, we did not include this figure in current

manuscript.

Figure not shown in manuscript: Representative western blot of CDK2 and Trim21 proteins in

different cancer cell lines.

The third figure is result of apoptosis induced by HHT in THP1 cells as suggested by the reviewer. We have now added the apoptosis results into the revised manuscript (Fig. S5B), as well as the description and implication of these results.

Fig. S5. The effects of CDK2 mutants on THP1 cell proliferation and response to HHT treatment. (A) Representative western blots of CDK2 protein in THP1 CDK2 knockout cells and THP1 CDK2-KO cells with re-expression of CDK2 or CDK2-3As. (B) Representative result of Annexin-V/DAPI staining assay to detect the apoptotic rate in THP1 and THP1CDK2^{-/-} cells with HHT treatment for 24 hours and the negative control with DMSO treatment. (C) Growth curves of THP1 CDK2 and THP1 CDK2-3As cells. (D) Representative western blots of CDK2 downstream target protein-NCL in THP1 CDK2-wt, CDK2-3As and CDK2-3As/T160E cells. (E) Cell survival curves of THP1 CDK2 and THP1 CDK2-3As cells with HHT treatment. Data represent the mean + SD from four independent biological samples for each group for (C and E). P values are indicated by two-tailed unpaired Student's t test for (C and E).

Reviewer #4 (Remarks to the Author):

In their manuscript, J. Zhang et al. search for new, non-ATP competitive chemical inhibitors of CDK2. Using an in silico screen, they identify homoharringtonine (HHT) as a candidate drug. The binding of HHT to CDK2, as well as the disruption of CDK2 binding to cyclin A & E, is validated by several in vitro assays. The specificity of HHT towards CDK2 and the efficacy of HHT in inhibiting cancer cell growth is validated using several genetically modified versions of CDK2 in cell culture assays, as well as by one in vivo assay. HHT's mechanisms of action if then studied in cultured cells, where the authors find evidence for HHT treatment resulting in CDK2 degradation. This degradation of CDK2 is shown to be autophagy-dependent and at least partially mediated by Trim21. Overall, the manuscript describes new and

interesting results, which could promote the study of CDK2 functions, suggest a new degradation mechanism for CDK2 and possibly even promote the development of new pharmaceuticals. If properly validated and carefully presented, these discoveries could warrant publication in Nature communications. Some experiments in the manuscript are very good, including the generation of CDK2-3As/T160E cell line to validate that HHT seems to act, at least partly, through CDK2.

However, in its current form the manuscript's work is completely unreproducible and a few key biological considerations have been ignored. Below, I have listed key issues, which should be addressed before publication in any high-quality journal could be considered. Notably, many of these can be addressed by writing. I have also listed more minor issues, which would help improve the quality and impact of the work, although they are less critical for accepting this work for publication.

Key issues for publication:

1. Homoharringtonine, similarly to its homolog harringtonine, is a translation inhibitor that block the elongation phase or ribosomes. The authors did not even mention this in the manuscript, despite the fact that this may explain many of the results authors observe. In cell-free assays even low nanomolar concentrations of HHT can inhibit translation (Selective inhibition of the polypeptide chain elongation in eukaryotic cells. 1992. Tujebajeva et al. PMID: 1730056), so it is critical for the authors to both discuss this in the manuscript and test if their discoveries are independent of any effects on protein synthesis. This could be done, for example, by comparing HHT's protein synthesis inhibition potential to HHT's potential in disrupting CDK2's binding to cyclin and to HHT's potential in stimulating CDK2 degradation. These experiments should be done carefully with several incubation times and model systems, including the CDK2-3As/T160E cell line in order to validate that protein synthesis inhibition is not due to HHT's effects on CDK2. If protein synthesis is not affected (or only very minimally affect) by the concentrations of HHT used in the study across multiple model systems, then the manuscript's findings are very interesting.

Response: Thanks for the critical comments. Previously, a report suggested that HHT could potentially inhibit protein synthesis by binding to the A site of ribosome. However, this effect could not explain all the anti-tumor effects elicited by HHT. In this study, we report that HHT could directly interact with CDK2 and induced its degradation in both *in vitro* and *in vivo* assays. More importantly, the results from the leukemia xenograft mouse model showed that the anti-leukemia effect of HHT was significantly attenuated when the CDK2 was knockout by CRISPR or CDK2 was mutated into CDK2-3As/T160E (Fig. S2 G&H).

Fig. 2. CDK2 is critical for AML proliferation. (G) Bioluminescence imaging of luciferase-expressing THP1 (CDK2-wt or CDK2-3As/T160E) xenograft mice models, treated with 0.5 mg/kg HHT (Six mice for each group). (H) Kaplan–Meier survival of luciferase-expressing THP1(CDK2-wt or CDK2-3As/T160E) xenograft mice models, treated with 0.5 mg/kg HHT or vehicle as the control. P values are calculated by the log-rank test.

As suggested, we also tested whether CDK2 status could affect protein synthesis. We determined the mRNA and protein levels of DDX5, STMN1 and MCL1 with protein half-life of ~9.6 h, ~3.7 h and ~30 min respectively, in THP1 cells with different CDK2 status. The results demonstrated that the mRNA and protein levels of DDX5, STMN1 and MCL1 did not change much in these cells (Fig. S9 A&B). To further determine whether CDK2 status may affect the general translation efficiency, the ribosome profiling was applied into these cells and the results showed that the absorbance peaks of the 40S and 60S ribosome subunits, as well as the monosome and polysome were similar among THP1 cells with different CDK2 status (Fig. S9C), suggesting that CDK2 knockout or CDK2 mutations did not affect the protein translation. Combined with the above results, targeting CDK2 may be an independent mechanism underlying the anti-leukemia effect of HHT, which is distinct from its potential effects on inhibiting protein synthesis as previously reported.

We have also included in the revised manuscript the discussion of these new results and the comparison to the protein translation inhibition by HHT as previously reported.

“Previous reports suggest this compound likely operates through a broad mode of action including protein synthesis inhibition and DNA epigenome modulation^{39, 40}. However, it is uncertain whether other undefined mechanisms underly the anti-tumor effects of HHT. Here we show that HHT can not only directly bind to the PPI site of CDK2 but also disrupt interactions between CDK2 and its cyclin partners. Moreover, this binding both inhibited the activity of

CDK2 and induced its degradation in cancer cells. Such findings thus highlight a potential therapeutic strategy to target CDK2-associated malignancies: the PPI between CDK2 and cyclins can be disrupted by a non-ATP competitive inhibitor of CDK2. Based on these findings, further studies are warranted to analyze the distinct functional groups in HHT molecule, which interact with specific target respectively to fulfill different activities including disruption of CDK2 PPI and protein translation blockage. Moreover, novel HHT derivatives could be

developed to inhibit different targets with higher specificity and lower requisite dosage with the analysis of functional groups in HHT molecule.”

Fig. S9. CDK2 status did not affect protein translation. The mRNA (A) and protein (B) level of DDX5, MCL1 and STMN1 in THP1 cells with different CDK2 status. Data represent the mean + SD from four independent biological samples for each group for (A). P values are indicated by one-way ANOVA with Tukey’s multiple comparison test for (A), and there is no statistical significance among these groups for each gene expression level. Source data are provided as a Source Data file. (C)The ribosome profiling of THP1 cells with different CDK2 status. The results shown here were representative of three independent experiments.

2. The Methods section is severely lacking to a point that the study could not be reproduced

at all. Without these corrections I would not recommend this manuscript for publication in any journal. Below is a list of missing details. Note that this list may not be comprehensive and the authors should make a serious effort in verifying that relevant details are presented.

Many chemical concentrations used, as well as some treatment times are still not reported (MG132, Roscovitine, NH₄Cl, etc.).

- Antibodies used and labeling details are not listed at all.
- LIVS method details are very superficial. More importantly, it is unclear if this is a new method, in which case the method section should be significantly more detailed, or if this is a previously established method, in which case references are missing.
- Pulldown experiment details are not presented.
- It is unclear where patient data was obtained from.
- siRNA details and usage information is completely missing.

Response: Thanks for all the insightful comments.

1. We have added the concentration and incubation time of each chemical used in the experiments in the manuscript and figure legends.

2. We have added the information of antibodies and qPCR primers in the method section as below: *“Western Blot Analysis. Cell specimens were washed twice with PBS buffer; total cellular protein was extracted using Radio-Immunoprecipitation Assay buffer (RIPA). Cell extracts were subjected to sodium dodecyl sulfate–polyacrylamide gel electrophoresis (SDS-PAGE; 10% polyacrylamide gels); then they were transferred to polyvinylidene difluoride (PVDF) membranes (Bio-Rad) and blocked with 5% nonfat milk (Bio-Rad) in TBS–Tween 20 (TBST). The membranes were then reacted with primary antibodies overnight at 4°C. After 3 washes with TBST, membranes were probed with a horseradish peroxidase–conjugated secondary antibody for 1h at room temperature and reacted with SuperSignal West Pico Chemiluminescent Substrate (Thermo). The antibodies were purchased from Proteintech as below: β -actin Monoclonal Antibody (#66009-1-Ig); HSP90 Polyclonal Antibody(#13171-1-AP); CDC37 Polyclonal Antibody(#10218-1-AP); ATG7 Polyclonal Antibody(#10088-2-AP); TRIM21 Polyclonal Antibody(#12108-1-AP); RUVBL2 Polyclonal Antibody(#10195-1-AP); DNAJA1 Polyclonal Antibody(#11713-1-AP); Beclin 1 Polyclonal Antibody(#11306-1-AP); GAPDH Monoclonal Antibody(#60004-1-Ig); HSP70 Polyclonal Antibody(#10995-1-AP); Cyclin A2 Polyclonal Antibody(#18202-1-AP); GST Tag Monoclonal Antibody(#66001-1-Ig); CDK1-Specific Polyclonal Antibody(#19532-1-AP). The antibodies were purchased from Cell Signaling Technology as below: DDX5 (D15E10) XP® Rabbit mAb(#9877); Stathmin Antibody(#3352); Mcl-1 (D5V5L) Rabbit mAb(#39224); Anti-mouse IgG, HRP-linked Antibody(#7076); Anti-rabbit IgG, HRP-linked Antibody(#7074); HA-Tag (C29F4) Rabbit mAb(#3724); CDK2 (E8J9T) XP® Rabbit mAb(#18048); Cyclin E1 (D7T3U) Rabbit mAb(#20808); Nucleolin (D4C7O) Rabbit mAb(#14574); Rb (4H1) Mouse mAb(#9303). Anti-Nucleolin (phospho T84) antibody (#ab155977) and Anti-Rb (phospho T821) antibody (#ab4787) were purchased from Abcam. Anti-HaloTag antibody (#G9281) and Monoclonal ANTI-FLAG® M2 antibody (#F1804) were purchased from Promega and Sigma respectively.”*

“Real-Time PCR. Total RNA (1 μ g), collected by the RNeasy kit (QIAGEN), was used in a reverse

transcriptase reaction with the SuperScript III Reverse Transcriptase kit (Life Technologies). The SYBR Green Real-Time PCR Master Mixes kit (Life Technologies) was used for the thermocycling reaction in an ABI-7500 RealTime PCR machine (Applied Biosystems). The real-time PCR analysis was carried out in triplicate with the following primer sets:

CDK2 Forward: 5'-CCAGGAGTTACTTCTATGCCTGA-3'

CDK2 Reverse: 5'-TTCATCCAGGGGAGGTACAAC-3'

DDX5 Forward: 5'-GCTTGCTGAAGATTCCTGAAAGAC-3'

DDX5 Reverse: 5'-TCTCACTCATGATCTCTCCATTAGAC-3'

Mcl-1 Forward: 5'-GGACATCAAAAACGAAGACG-3'

Mcl-1 Reverse: 5'-GCAGCTTTCTTGGTTTATGG-3'

STMN1 Forward: 5'-GCCCTCGGTCAAAAAGAATCTG-3'

STMN1 Reverse: 5'-TGCTTCAAGACCTCAGCTTCA-3'

GAPDH Forward: 5'-CCACATCGCTCAGACACCAT-3'

GAPDH Reverse: 5'-GCGCCCAATACGACCAAAT-3''''

3. We have added 4 references (a-d) for the descriptions of the LIVS pipeline, along with the inhibitors resulted from its applications. LIVS is our in-house developed pipeline to perform virtual ligand screening. It is a unique method that has been implemented to find more than 10 potent inhibitors in nano-molar or micro-molar range.

References:

- a. Rui Su, Lei Dong, Yangchan Li, Min Gao, Li Han, Mark Wunderlich, Xiaolan Deng, Hongzhi Li, Yue Huang, Lei Gao, Chenving Li, Zhicong Zhao, Sean Robinson, Brandon Tan, Ying Qing, Xi Qin, Emily Prince, Jun Xie, Haniun Qin, Wei Li, Chao Shen, Jie Sun, Prakash Kulkarni, Hengyou Weng, Huilin Huang, Zhenhua Chen, Bin Zhang, Xiwei Wu, Mark I Olsen, Markus Müschen, Guido Marcucci, Ravi Salgia, Ling Li, Amir T Fathi, Zejuan Li, James C Mullov, Minjie Wei, David Horne, Jianiun Chen, "Targeting FTO suppresses cancer stem cell maintenance and immune evasion", *Cancer cell*, 2020, 38(1), 79-96
- b. Ian Tian, Min-Sung Kim, Hongzhi Li, Jimin Wang, and Wei Yang "Structure of HIV-1 reverse transcriptase cleaving RNA in an RNA/DNA hybrid", *Proceedings of the National Academy of Sciences*, vol 115, 507-512, 2018
- c. Liu X, Xie R, Yu H, Chen SH, Yang X, Singh AK, Li H, Wu C, Yu X. A126 inhibits the ADP-ribosylhydrolase ARH3 and suppresses DNA damage repair. *J Biol Chem*. 2020 Oct 2;295(40):13838-13849
- d. Wenneng Liu, Mian Zhou, Zhengke Li, Hongzhi Li, Piotr Polaczek, Huifang Dai, Qiong Wu, Changwei Liu, Kenneth K. Karania, Venkat Ponuri, Shu-ou Shan, Katharina Schlacher, Li Zheng, Judith I. Campbell, Binghui Shen. A selective small molecule DNA2 inhibitor for sensitization of human cancer cells to chemotherapy, *EBioMedicine*, 2016, Vol.6, p73-86

4. We have added the description of pulldown assay in the method section as below:
"Pulldown assay by HHT-conjugated magnetic beads. The HHT-conjugated magnetic beads were synthesized as below: one ml of Absolute Mag™ Epoxy Magnetic Particles (#WHM-Q009, CD Bioparticles) was washed with dimethylformamide (DMF) for three times and suspended in 5 ml of DMF. Seventy-five mg of HHT was added into mixture with 200 µL of triethylamine (TEA), and the mixture was stirred overnight at room temperature protected from light. The magnetic beads were washed by PBS for 5 times to remove unreacted residue and resuspended in 1 ml of TBS. After conjugation, fourier transform infrared spectroscopy (FTIR) was used to check the binding between the epoxy of the beads and HHT. HEK293 cells were lysed in M-PER buffer with the addition of both protease inhibitors and phosphatase inhibitors. One hundred µL of HHT-conjugated magnetic beads and control magnetic beads were added into the protein lysate separately, and the mixture was rotated at 4°C overnight. Then the magnetic beads were

washed by M-PER buffer for 5 times and the proteins binding to the HHT-conjugated magnetic beads were harvested by heating the samples in 60 μ L of SDS loading buffer at 95°C for 10 min.”

5. Patient data were obtained from the peripheral blood samples of patients with AML and normal volunteers with their written informed consent in The Second Affiliated Hospital of Zhejiang University School of Medicine. We have described the information in the method section as below: “Peripheral Blood Mononuclear cells (PBMC) Isolation. Mononuclear cells were isolated using lymphocyte separation medium from the peripheral blood samples of patients with AML and normal volunteers with their written informed consent. Briefly, blood was diluted 1:1 in phosphate-buffered saline (PBS) at RT prior to layering over Lymphocyte Separation Medium. PBMC were collected following centrifugation (800g, 20 min) and washed in PBS (320g, 10 min). Cells were resuspended in RPMI-1640 medium supplemented with 10% fetal bovine serum. This study was carried out in accordance with the recommendations of Ethics and Scientific Committee of The Second Affiliated Hospital of Zhejiang University School of Medicine with written informed consent from all subjects.”

6. We have added the siRNA transfection information into the method section as below: “siRNA transfection. The transfection was following Lonza’s protocol with Nucleofector I Device. One million cells were harvested and centrifuged at 90xg for 10 minutes at room temperature. Resuspend the cell pellet carefully in 100 μ l room-temperature Nucleofector Solution per sample. Combine 100 μ l of cell suspension with 300 nM siRNA. Transfer cell/DNA suspension into certified cuvette and insert the cuvette with cell/DNA suspension into the Nucleofector Cuvette Holder. Select the appropriate Nucleofector Program U-01 and apply the selected program. Take the cuvette out of the holder once the program is finished. Immediately add 500 μ l of the pre-equilibrated culture medium to the cuvette and gently transfer the sample into the prepared 12-well plate. Incubate the cells in humidified 37°C/5% CO₂ incubator until analysis. The siRNAs for ATG7 and Trim21 are SMART pool from Dharmacon: ATG7 (#L-020112-00) and Trim21 (#L-006563-00). The siRNA for GFP is 5’-CAGAUGAACUUCAGGGUCAGC-3’.”

As suggested, all other relevant details were added as necessary in this revised manuscript.

3. Reproducibility of most results is unclear. How many times were each experiment repeated? Optimally, the authors would show quantifications of all replicates for western blots next to the actual blots, especially for key figures.

Response: Thanks for the comments. Each experiment was repeated for 3 times. According to the arrangement of each figure, we put the relative quantification of each band under respective band in western blots.

4. Experiments in figure 2G and 2H do not proof that HHT has to act through CDK2 in vivo, because the CDK2 KO model had almost influence on animal survival in the first place. Thus, it remains unclear if HHT would have also increased survival in the CDK2 KO model, especially as the experiment relied on very few mice. Optimally, this would have been done using the CDK2-3As/T160E model, with longer-term examination of the mice and

with larger replicate numbers.

Response: Thanks for this critical comment. We have repeated the *in vivo* experiment with CDK2-3As/T160E model and analyzed the effects of HHT on leukemia development by luciferase intensity and mice survival rate in larger replicate numbers (n=6). At the end of this experiment, all THP1 CDK2-wt xenograft mice with vehicle treatment died while THP1 CDK2-

wt xenograft mice with HHT treatment were all alive. There were 3 mice died in the group of THP1 CDK2-3As/T160E xenograft mice with vehicle treatment and 2 in the group treated with HHT (Fig 2G and H). The experiment demonstrated that HHT treatment inhibited the leukemia development in THP1 CDK2-wt xenograft mice but had limited effect on THP1 CDK2-3As/T160E xenograft.

Fig. 2. CDK2 is critical for AML proliferation. (G) Bioluminescence imaging of luciferase-expressing THP1 (CDK2-wt or CDK2-3As/T160E) xenograft mice models, treated with 0.5 mg/kg HHT (Six mice for each group). (H) Kaplan–Meier survival of luciferase-expressing THP1(CDK2-wt or CDK2-3As/T160E) xenograft mice models, treated with 0.5 mg/kg HHT or vehicle as the control. P values are calculated by the log-rank test.

Minor points to improve the work:

1. The manuscript could use proof reading and careful validation of references. For example, the abstract misspells 'cyclin' as 'cylcin', and when discussing "aberrant activation of CDK2" (line 49 & 50 in intro) the authors reference ref #9 which has no data on any 'aberrant activation of CDK2'.

Response: Thanks for the comments. We have made the proof reading and corrected the misspells. Also, we have replaced the ref #9 with the right reference.

Ref^{#9}: Tadesse S, et al. Targeting CDK2 in cancer: challenges and opportunities for therapy. *Drug discovery today* 25, 406-413 (2020).

2. Some of the claims in the manuscript need to be dialed back, especially for non-specialist audiences. In silico predictions of drug binding should be discussed as predictions until comprehensive experimental validations are carried out.

Response: As suggested, we use “predict” instead of “identify” for in silico predictions of drug binding.

3. The authors validated that CDK1 does not seem to be affected by HHT. However, the CDK2 phosphorylation targets studied are also phosphorylated by the closely related CDK4. The results could be influenced by HHT inhibiting CDK4. This should be preferably tested or at least acknowledged in writing.

Response: Thanks for the suggestion. Because CDK1 shares the most similarity with CDK2, we choose CDK1 as the representative to show that, compared to other CDKs, the binding of HHT to CDK2 is more specific and the interaction of HHT with CDK2 is stronger: *“CDK2 promotes cell cycle progression cooperating with CDK1, CDK4/CDK6 and respective cyclins. Among these 4 CDK molecules, CDK2 protein shares the highest similarity with CDK1 protein.”*

4. Regarding data in figure 4 A-D, the authors state that “the ability of HHT to effectively inhibit the development of leukemia in a mouse model appeared to involve the degradation of CDK2 protein.” (line 208). Technically, no causality between development of leukemia and CDK2 degradation is established in these experiments and the conclusion is therefore an overstatement.

Response: Thanks for the comments. We have changed the conclusion as: *“the ability of HHT to effectively inhibit the development of leukemia in a mouse model may be relevant to the degradation of CDK2 protein.”*

5. Experiments in figure 5C. MG132 and NH4Cl are recognized as somewhat dirty drugs with many off-targets. These experiments would be more convincing if the authors repeated them with other proteasome and autophagy inhibitors.

Response: Thanks for the suggestion. We have further tested lactacystin as the proteasome inhibitor and chloroquine as the autophagy inhibitors and we observed the similar results: *“To exclude the off-targeting possibility of these two inhibitors, we introduced two more inhibitors-Chloroquine (an autophagy inhibitor, 50 μ M) and lactacystin (a proteasome inhibitor, 10 μ M) to further confirm that the blockage of autophagy pathway but not proteasome pathway could reverse the effect of HHT on the degradation of CDK2 protein (Fig 5C lower panel).”*

Fig. 5. Identification of Trim21 as a mediator of CDK2 autophagic degradation induced by HHT treatment. (C) Western blot analysis in THP-1 cells were pretreated with NH₄Cl (20 mM) or MG132 (5 μ M) (upper panel) and Chloroquine (50 μ M) or lactacystin (10 μ M) (lower panel), together with HHT treatment for 9 hours.

6. On lines 253 and 254 the authors discuss the discoveries of their MS/MS experiment. Where is the data?

Response: Thanks for this comment. The MS data is added as the Supplementary Dataset in this revised manuscript.

7. Figure S9B requires quantifications and controls (wt cells) should also be shown next to the two mutants.

Response: As suggested, we have added the quantifications and controls in Figure S11, which was the previous Figure S9.

Fig. S11. (B) Pulse-chase analysis of different CDK2 mutants using HT-TMR system in 293T cells transfected with HT-CDK2-wt (left panel), HT-CDK2- Δ 171-243 (middle panel) or HT-CDK2-5Rs (right panel). The HT-TMR ligand-labeled HT-CDK2 was visualized with a fluoro-image analyzer after the treatment with HHT at 100 ng/mL. The total level of HT-CDK2-5Rs and HT-CDK2- Δ 171-243 were determined by western blot with GAPDH as the loading control.

8. The microscopy in the last main figure is extremely limited in its usefulness, as no replicates are shown, no methods are listed and no quantifications are done. Essentially, the manuscript gains nothing from these images, although more careful investigation could be interesting. Otherwise, even removal of these images could be a better option.

Response: As suggested, we added the statistical analysis of the colocalization of CDK2 and Trim21: "As shown in Fig 6E upper panel, the treatment both strongly reduced the levels of CDK2-GFP and induced co-localization between CDK2-GFP and mRuby2-Trim21; these effects were followed by autophagic degradation of CDK2. In contrast, a similar cascade was not observed for CDK2-3As-GFP. We checked the colocalization between CDK2-GFP and mRuby2-Trim21 induced by HHT in three different replicates, and the results showed significant

colocalization in CDK2-GFP group but not in CDK2-3As-GFP group after HHT treatment (Fig 6E lower panel).”

Fig.6. (E) HEK293 cells were transfected with either CDK2-GFP or CDK2-3As-GFP as well as mRuby2-Trim21; then cells were either treated with HHT (50 ng/mL) or DMSO for 9 hours, and the localizations of CDK2 (green) and Trim21 (red) were determined by fluorescence microscopy (upper panel). Scale bar, 20 μ m. The positive cell ratio with colocalization between CDK2-GFP and mRuby2-Trim21 after HHT treatment in three different experiment replicates was analyzed (lower panel). Data represent the mean \pm SD from three independent biological samples for each group. P values are indicated by two-tailed unpaired Student's t test.

- The discussion claims that “When CDK2 inhibitors are added to a range of established regimens, the inhibitors not only improve the responses of some cancer patients but also overcome other patients’ resistance to therapy.” (line 322) This requires references. Furthermore, this is soon followed by statement that “The vast majority of CDK2 inhibitors target the ATP binding site of CDK2. ...These inhibitors failed to meet expectations during clinical trials because they have either poor specificity or high toxicity.” This seems very contradictory to the previous sentence, especially in the absence of references.

Response: Thanks for the comments. We have corrected the statement as below:

“Dysregulation of CDK2 and its complex plays a critical role in both the malignant transformation of cells and tumorigenesis. When CDK2 inhibitors are applied to a range of cancer cells in vitro and tumor mice models in vivo, the inhibitors exhibit an encouragingly anti-tumor effect. However, their utility within a clinical setting has been impeded largely due to the lack of requisite specificity for CDK2.”

- The discussion states that “While other computational models typically produce a hit rate of approximately 3%, the hit rate for LIVS is nearly 15%.” (line 341). This is not clear in the results section and it seems that the authors are discussing the number of predictions of made by LIVS. Yet, having more predictions is not useful if they are incorrect, as seems to be the case for most predictions in this study, as shown in figure S1. Thus, this sentence seems very misleading, unless the authors specify that they are discussing the ‘non-validated hit rate’.

Response: We apologize that we did not make it clear in manuscript that the hit rate is defined as percentage of virtual ligand screening compounds that passed the primary assay in wet-lab validation with a typical concentration of 20-50 μ M. This rate can be used to estimate the methodology efficiency among different virtual screening methods. For example, in our FTO inhibitor development (Cancer cell. 2020, 38, 79-96), a list of 213 virtual screening candidates

were generated by LIVS method and 40 compounds with concentration $\leq 50\mu\text{M}$ passed the initial cell viability in MONOMAC 6 assay, resulting in a hit rate of $40/213=18.8\%$. In this manuscript, we only generated 10 virtual screening compounds for experimental validation, thus the hit-rate is 10%. We agree that more predictions are not useful if they are incorrect. The number of predicted compounds is carefully controlled by LIVS pipeline based on the factors such as the size of compound library, experimental capacity etc. We added the definition of hit rate in manuscript, as well as two more references.

11. The discussion is largely a listing of the results of the manuscript. This seems like a missed opportunity for the authors to discuss the larger scale implications of their work. Could HHT be easily modified to increase its specificity to CDK2? Which other diseases might benefit from CDK2 inhibition? Does the autophagic degradation of CDK2 suggest that other CDKs are likely to be degraded by autophagy too? Etc...

Response: Thanks for the comments on the discussion part. We have briefly summarized the results in the Discussion and spent more effort to discuss the larger scale implications of our results, including the application of HHT in different cancers and the future directions to modify HHT.

Reviewer #5 (Remarks to the Author):

Zhang et al. propose that HHT, a known therapeutical, is interacting with CDK2 to disrupt Cyclin A binding and thus inactivates CDK2. Moreover, they propose that binding of HHT to CDK2 induces its degradation via an autophagy lysosomal pathway. They could show in addition that Trim21, a protein involved in the aforementioned pathway interacts with CDK2 in a way that is mutually exclusive with CyclinA binding. Overall, this a well conducted study that opens interesting new possibilities in the repurposing of HHT.

However, in addition to the previous reviewers comments some issues need to be resolved prior to publication.

Although, the methodology how the authors identified HHT as a possible CDK2 binding compound is sound and seems plausible the follow up characterization of the mechanism of action still lacks some experimental conformation.

1. The authors propose that HHT binds to a certain area on the surface with a predicted hydrogen bonding pattern. To interrupt binding they generated a variant that contains alanine exchanges for residues T47, R50 and R150. In pulldown experiments they show that this variant cannot bind to HHT beads anymore suggesting a disrupted interaction. It is, however, unclear whether this lack of interaction occurs due to the disruption of binding to HHT via the proposed interface or whether this variant is properly folded. A closer look at the structure suggests that this triple variant might significantly influence the local fold. The authors should thus perform comparative folding studies to ensure structural integrity of the variant. This could be accomplished for example either by CD spectroscopy or Thermofluor studies. This analysis should be also accompanied by in vitro kinase/ATPase assays with the recombinant protein in the presence and absence of Cyclin A and HHT. In addition to demonstrating that the fold is maintained, the kinase/ATPase

analysis would also yield direct evidence of the mechanism of action which would strengthen their point that HHT disrupts the Cyclin A interface.

Response: Thanks for the insightful comments. As suggested, we have performed the nanoDSF assay. The results showed that CDK2 protein and its mutants had similar melting curve, suggesting the mutations did not significantly affect the CDK2 protein structure. Fig. S6A showed the DSF profiles of wild-type CDK2, CDK2-3As and CDK2-3As/T160E proteins, all of which exhibited similar nanoDSF traces. The apparent T_m of wild-type CDK2 is $\sim 51^\circ\text{C}$, and the mutants CDK2-3As and CDK2-3As/T160E showed similar thermal stability within $\sim 0.2^\circ\text{C}$ difference as compared to the wild-type CDK2. This experiment suggested that mutations of residues T47, R50 and R150 did not significantly affect the folding of CDK2 protein. As suggested, we also compared the kinase activities among CDK2, CDK2-3As and CDK2-3As/T160E by CDK2 kinase assay (Fig. S6B). With the addition of Cyclin A2, the kinase activity of CDK2-3As or CDK2-3As/T160E was significantly lower than that of wild-type CDK2. The relative activity of CDK2-3As was around 45% and CDK2-3As/T160E around 80% as compared to that of wild-type CDK2. When HHT was introduced, the activity of wild-type CDK2 was significantly impaired, while the activity of CDK2-3As or CDK2-3As/T160E did not change much. These results suggest that the HHT may inhibit the *in vitro* kinase activity of CDK2 via interaction with the residues T47, R50 and R150 of CDK2.

Fig S6. NanoDSF and Kinase activity analysis of CDK2 and its mutants. (A) nanoDSF traces with the relative first derivatives for CDK2 and the mutants. X-axis represents temperature in $^\circ\text{C}$ and y-axis represents first derivative of ratio of intrinsic fluorescence (350:330 nm). The black, red, blue and green solid lines show the curves of wild-CDK2, CDK2-3As, CDK2-3As/T160E and CDK2-5Rs measured in assay buffer (150 mM NaCl, 100 mM Tris, pH 7.5). Three replicates for each sample. (B) The kinase activity assay for wild-CDK2, CDK2-3As and CDK2-3As/T160E protein with the presence of cyclin A2 (20 ng) and HHT (100 ng/mL) or not. Data represent the mean + SD from four independent biological samples for each group. P values are indicated by one-way ANOVA with Tukey's multiple comparison test. Source data are provided as a Source Data file.

- The results represented in the manuscript are a combination of HHT resin association of CDK2 and pull-down experiments that have been performed in cell based experiments.

They show a clear tendency towards the proposed mechanism of action concerning HHT but do not unambiguously prove it. The IC₅₀ values of the suggested experiments would also put the *in vivo* IC₅₀ values into context that have been obtained with the MTS assay.

Response: Thanks for the insightful comment. Our in-house LIVS method, which have been successfully applied in the previous studies (Cancer cell. 2020, 38, 79-96), predicted a direct binding between HHT and CDK2 proteins. Pulldown assay and co-IP assays with different CDK2 mutants confirmed this interaction and the binding of HHT to CDK2 disrupted the interaction between CDK2 and its partner—Cyclin A/Cyclin E, which induced the degradation of CDK2 protein in both *in vitro* and *in vivo* assays. We have also tried the co-crystallization experiment to demonstrate a direct binding of HHT to CDK2. However, the low solubility of HHT in water impeded the success of this experiment, which requires further endeavor in the future.

Also, as suggested, we have added the *in vivo* IC₅₀ value in the Fig.4G.

Fig. 4. Induction of CDK2 degradation *in vivo*. (G) The correlation between CDK2 expression and IC₅₀ of HHT in human primary AML specimens (E). Correlation is shown using r^2 and significance was determined using a Spearman correlation. The numbers close to the dots are the IC₅₀ of HHT for respective sample (ng/mL).

3. With respect to the T160E variant of the CDK2-3As variant it is entirely unclear how this variant fully reactivates CDK2 activity. Aren't both T-loop phosphorylation and cyclin binding both equally important to reach full activity for CDKs? How this variant is able to compensate the previous phenotype of the CDK2-3As is elusive at this point and should be explained more thoroughly, accompanied by the already proposed *in vitro* kinase/ATPase studies.

Response: Thanks for this insightful comment. Figure 2E showed that the proliferation of CDK2-3As/T160E cells was delayed in 3 days as compared with that of wild type cells, and figure 2G and 2H further demonstrated that the leukemia development of CDK2-3As/T160E was slower compared to that of wild-type CDK2, indicating an impaired activity of CDK2-3As/T160E in vivo. Both experiments suggested that the CDK2-3As/T160E still displays some defects, thus could not replace the wild type CDK2 completely. We have made the correction

accordingly in the Text.

Fig. 2. CDK2 is critical for AML proliferation. (D) Cell survival curve of THP1 wild-type and THP1 CDK2 knockout cells after HHT treatment for 24 hours. (E) Growth curves of THP1 CDK2 and THP1 CDK2-3As/T160E cells. (F) Cell survival curve of THP1 CDK2 and THP1 CDK2-3As/T160E cells after HHT treatment for 24 hours. Data presented as mean \pm SD from four independent biological samples for each group for (C, D, E and F). P values are indicated by two-tailed unpaired Student's t test for (C, D, E and F). (G) Bioluminescence imaging of luciferase-expressing THP1 (CDK2-wt or CDK2-3As/T160E) xenograft mice models, treated with 0.5 mg/kg HHT (Six mice for each group). (H) Kaplan–Meier survival of luciferase-expressing THP1(CDK2-wt or CDK2-3As/T160E) xenograft mice models, treated with 0.5 mg/kg HHT or vehicle as the control. P values are calculated by the log-rank test. Source data are provided as a Source data file.

In addition, we compared the kinase activities among CDK2, CDK2-3As and CDK2-3As/T160E by CDK2 kinase assay (Fig. S6B). With the addition of Cyclin A2, the kinase activity of CDK2-3As or CDK2-3As/T160E was lower than that of wild-type CDK2 significantly. The relative activity of CDK2-3As was around 45% and CDK2-3As/T160E was around 80% as compared to wild-type

CDK2. When HHT was added, the activity of wild-type CDK2 was significantly impaired, while the activity of CDK2-3As or CDK2-3As/T160E did not change much. These results indicate that the HHT may inhibit the *in vitro* kinase activity of CDK2 via interaction with the residues T47, R50 and R150 of CDK2.

Fig S6. Kinase activity analysis of CDK2 and its mutants. (B) The kinase activity assay for wild-CDK2, CDK2-3As and CDK2-3As/T160E protein with the presence of cyclin A2 (20 ng) and HHT (100 ng/mL) or not. Data represent the mean + SD from four independent biological samples for each group. P values are indicated by one-way ANOVA with Tukey's multiple comparison test. Source data are provided as a Source Data file.

- Another concern are the experiments concerning the CDK2 Δ 171-243 variant. It is most likely that the resulting protein is not suitable for any meaningful interpretation since a significant portion of the C-terminal kinase lobe is missing which almost certainly disrupts the folding of CDK2. These results should therefore be interpreted with extreme caution. In addition, the authors propose that the region V154/P155/M233/P234/P238 could form hydrogen-bonds to Trim24, which is most likely a typo and should be corrected. To investigate this in detail a 5R variant (V154R/P155R/M233R/P234R/P238R) of this particular region was constructed to disrupt this interaction. Since none of these residues can undergo side chain hydrogen bonding interactions it must be assumed that the authors only observed main chain interactions with CDK2 at these positions? These are notoriously difficult to tackle and hence they have chosen a drastic approach by changing all residues to an arginine. The three prolines in this region could be of structural importance and again mutations could severely affect the fold in this region rendering their results less interpretable. Here again the analysis of proper folding as recommended for the other variants seems to be essential.

Response: We appreciate this insightful comment. Indeed, the deletion of 171-243 amino acid residues in CDK2 might disturb the folding of CDK2 protein, leading to loss of its interaction with Trim21. Therefore, we change the statement as: "And the results demonstrated that deletion of 171-243 amino acid residues in CDK2 protein might abolish the interaction between CDK2 and Trim21 induced by HHT (Fig S11A)." To exclude this possibility, we took an alternative approach to introduce point mutations in CDK2 protein (CDK2-5Rs: V154R/P155R/M233R/P234R/P238R mutations). The NanoDSF assay showed that CDK2-5Rs protein had a similar melting curve with wild-type CDK2 protein (Fig.S6A), indicating that the mutated CDK2-5Rs protein folded properly. The results of Co-IP experiment demonstrated that interaction of CDK2-5Rs with Trim21 could not induced by HHT, while that of wild-type CDK2 could (Fig. 5K). Therefore, there results suggested that CDK2-5Rs disrupted the hydrophobic interactions between the two proteins, thereby abolishing the interaction between CDK2 and

Trim21 induced by HHT.

Fig S6. NanoDSF and Kinase activity analysis of CDK2 and its mutants. (A) nanoDSF traces with the relative first derivatives for CDK2 and the mutants. X-axis represents temperature in °C and y-axis represents first derivative of ratio of intrinsic fluorescence (350:330 nm). The black, red, blue and green solid lines show the curves of wild-CDK2, CDK2-3As, CDK2-3As/T160E and CDK2-5Rs measured in assay buffer (150 mM NaCl, 100 mM Tris, pH 7.5). Three replicates for each sample.

K

Fig. 5. Identification of Trim21 as a mediator of CDK2 autophagic degradation induced by HHT treatment. (K) HEK293 cells transfected with Trim21-HA plasmid and 3xFLAG-CDK2 or 3xFLAG-CDK2-5Rs plasmid were treated with HHT (50 ng/mL) for 3 hours, followed by co-immunoprecipitation with HA-beads and western blot analysis.

We apologize for the typo to say that “the region V154/P155/M233/P234/P238 could form hydrogen-bonds to Trim24”. It is corrected as: “hydrophobic interactions”. We selected to mutate them to Args together for two reasons. First, based on the model, protein-protein interface of CDK2/Trim24 are mainly hydrophobic interactions from 5 pairs as V154(CDK2)/P114(Trim24), P155/W100, M233/F167, P234/F167 and P238/W16. Second, although hydrophobic interactions are essential for many protein-protein bindings, individual hydrophobic interaction pair itself usually is not strong. Therefore, we mutated all the 5 residues in CDK2 to hydrophilic and long arginine in order to block the hydrophobic interactions between the two proteins. It is correct that the proline is a special residue that could change protein structure, especially in α -helices and sharp turns. However, as shown in the figure below, the 3 prolines are in the relatively flexible loop region of CDK2, which may have less effect on CDK2 structure when mutated. Moreover, The NanoDSF assay as mentioned above showed that CDK2-5Rs protein has a similar melting curve with wild-type CDK2 protein (Fig.S6A), indicating that the mutated CDK2-5Rs protein can fold properly.

5. It would be helpful if the authors could indicate the HHT binding region in Figure 5J of the model, since it should not interfere with the Trim21 binding region.

Response: As suggested, we have indicated the HHT binding region in Figure 5J of the model.

Fig. 5. Identification of Trim21 as a mediator of CDK2 autophagic degradation induced by HHT treatment. (J) The complex model of Trim21 and CDK2 proteins. The protein-protein interaction residues on Trim21 (green-colored cartoons) C-terminal region to protein IGHG1 (PDB -code: 2IWG) are displayed as blue-colored sticks. The cyclin A protein is also superimposed as grey cartoons (PDB -code: 1FIN) to show its overlap with Trim21. The model of HHT bound to CDK2 is superimposed as pink sticks.

REVIEWERS' COMMENTS:

Reviewer #2 (Remarks to the Author):

The authors have addressed the concerns raised in the previous review.

Response: Thank you for the support.

Reviewer #4 (Remarks to the Author):

The authors have done extensive revisions to their manuscript, including major experimental work and rewriting. These revisions have clearly improved the quality and reproducibility of their work. Overall, the authors have sufficiently addressed all my feedback and I support the publication of this manuscript.

Response: Thank you for the support.

Reviewer #5 (Remarks to the Author):

The authors have addressed all the issues and I support publication of the article.

One small comment regarding the methods: Please include the fluorescent dye used in the nano differential scanning fluorimetry method.

Response: Thank you for the support. And the NanoDSF is a dye-free DSF method that monitors the change of intrinsic fluorescence from inherent tryptophan in protein as a function of temperature, time, or denaturant concentration.